# Identification of fungal lignocellulose-degrading biocatalysts secreted by *Phanerochaete chrysosporium* via activity-based protein profiling

Christian Schmerling[1,8], Leonard Sewald [2,8], Geronimo Heilmann [2,7,8], Frederick Witfeld[3], Dominik Begerow [3], Kenneth Jensen [4], Christopher Bräsen[1], Farnusch Kaschani [2,5], Herman S. Overkleeft[6], Bettina Siebers [1✉] & Markus Kaiser [2✉]

Activity-based protein profiling (ABPP) has emerged as a versatile biochemical method for studying enzyme activity under various physiological conditions, with applications so far mainly in biomedicine. Here, we show the potential of ABPP in the discovery of biocatalysts from the thermophilic and lignocellulose-degrading white rot fungus *Phanerochaete chrysosporium*. By employing a comparative ABPP-based functional screen, including a direct profiling of wood substrate-bound enzymes, we identify those lignocellulose-degrading carbohydrate esterase (CE1 and CE15) and glycoside hydrolase (GH3, GH5, GH16, GH17, GH18, GH25, GH30, GH74 and GH79) enzymes specifically active in presence of the substrate. As expression of fungal enzymes remains challenging, our ABPP-mediated approach represents a preselection procedure for focusing experimental efforts on the most promising biocatalysts. Furthermore, this approach may also allow the functional annotation of domains-of-unknown functions (DUFs). The ABPP-based biocatalyst screening described here may thus allow the identification of active enzymes in a process of interest and the elucidation of novel biocatalysts that share no sequence similarity to known counterparts.

[1] Molecular Enzyme Technology and Biochemistry (MEB), Environmental Microbiology and Biotechnology (EMB), Centre for Water and Environmental Research (CWE), Faculty of Chemistry, University of Duisburg-Essen, Universitätsstraße 5, 45141 Essen, Germany. [2] Department of Chemical Biology, ZMB, Faculty of Biology, University of Duisburg-Essen, Universitätsstraße 2, 45117 Essen, Germany. [3] Evolution of Plants and Fungi, Ruhr-University Bochum, Universitätsstraße 150, 44780 Bochum, Germany. [4] Novozymes, Biologiens vej 2, 2800 Kgs, Lyngby, Denmark. [5] Analytics Core Facility Essen, ZMB, Faculty of Biology, University of Duisburg-Essen, Universitätsstraße 2, 45117 Essen, Germany. [6] Department of Bio-organic Synthesis, Leiden Institute of Chemistry, Leiden University, Einsteinweg 55, 2333 CC Leiden, Netherlands. [7] Present address: German Diabetes Center (DDZ), Leibniz Center for Diabetes Research at Heinrich Heine University Düsseldorf, Düsseldorf, Germany. [8] These authors contributed equally: Christian Schmerling, Leonard Sewald, Geronimo Heilmann. ✉email: bettina.siebers@uni-due.de; markus.kaiser@uni-due.de

Activity-based protein profiling (ABPP) has emerged as a widely used chemical proteomics methodology for basic biology research[1–5]. In ABPP, activity-based probes (ABPs) consisting of a reactive "warhead", an enzyme inhibitory moiety that forms a covalent irreversible bond with their target protein(s) and often ensures a high enzyme class specificity, a linker, and a reporter tag are used to label, identify and report on active enzymes under native, physiological conditions. As reporters, biotin, fluorophores, or so-called two-step reporter tags such as alkyne or azide moieties are frequently used[6]. In the last years, intensive efforts have been undertaken, both to develop new ABPs targeting new enzyme families and to establish their use in, amongst others, drug discovery (target and lead discovery, target engagement)[7–12], plant biology[13], or microbiology[14–18].

An evolving alternative ABPP application is its usage in biocatalyst screening and, thus, biotechnology (we here define biocatalysts as enzymes with potential industrial usage)[19–21]. For example, ABPP can be used to uncover microbial biocatalysts able to turn over complex polymeric biomass like lignocellulose, entities much sought in the context of sustainable energy and circular bioeconomy processes[22–24]. In contrast to sequence homology-based biocatalyst screening approaches (for instance, the analysis of genomics data[25]), ABPP exploits the established enzyme selectivity of ABPs to identify new biocatalysts with desirable substrate preferences, in principle, without the need to assign sequence homologies (Fig. 1a)[26,27]. In what is termed "ABP-based enrichment", only those enzymes that are active and thus have the capability to react with an ABP are selected for ensuing identification by LC-MS/MS-based sequencing, which limits protein identification according to the target specificity of the used ABPs. Accordingly, the often-cumbersome biochemical protein expression and purification is limited to only those biocatalysts preselected by ABPP—an enormous reduction of work when compared to screening methods that rely on systematic protein expressions. This is particularly relevant in fungal biocatalyst screen campaigns, which are frequently hampered by difficult heterologous protein expression and purification, e.g., as a result of complex glycosylation patterns in the case of wood-degrading enzymes[28,29]. Despite these intrinsic advances, ABPP-based biocatalyst discovery has been applied mostly in proof-of-concept studies using pure cultures, often of model organisms, and, more importantly, of standard culture media for their growth[30–34].

In the present study, we aimed to overcome this limitation and showcase the potential of the ABPP biocatalyst screening technology in a more complex and biotechnologically-relevant experimental setting. Accordingly, we applied ABPP to a suspended culture consisting of the white rot fungus *Phanerochaete chrysosporium* grown on minimal medium and solid beech wood chips as the sole carbon and energy source, similar to recent work regarding a functional ABPP approach in a set of basidiomycetes[35]. Lignocellulose is the main component of dead wood and represents a highly recalcitrant polymeric complex built up from cellulose, xylan (hemicellulose), and lignin (Fig. 1b)[36]. Sustainable methods for its efficient degradation are urgently sought for establishing a biotechnological conversion of non-food biomass into an industrial feedstock[37–39]. This, however, requires a synergistic action of different biocatalysts such as glycoside hydrolases (GHs), carbohydrate esterases (CEs), polysaccharide lyases, and other enzymes belonging to the auxiliary enzyme class (auxiliary activities, AAs), such as lytic polysaccharide monooxygenases (LPMOs)[40–42]. *P. chrysosporium* is an effective degrader of dead wood and lignocellulose in particular[43]. Its genome harbors a large repertoire of lignocellulolytic enzymes consisting of more than 69 different carbohydrate-active enzyme (CAZyme) families, including a total of 166 GHs, 14 CEs, and 57 glycosyltransferases (GT) (as listed in the CAZY database (www.cazy.org[44])[45]. This enormous complexity turns this organism into a promising resource for biocatalyst discovery. However, due to its sheer size, the *P. chrysosporium* secretome cannot be explored by systematically expressing all enzymes. Instead, a methodology to preselect only those enzymes directly involved in lignocellulose degradation is required. Of note, the expression of these biocatalysts are regulated by the presence of the lignocellulosic substrate, demanding preselection assays in the presence of insoluble wood chips and thus highly heterogenous conditions[46].

With the aim to rapidly identify promising lignocellulose-degrading enzymes from this complex system via ABPP-based preselection, we applied FP-alkyne, a well-established serine hydrolase (SH)-targeting ABP[47,48], and the two structurally-related GH-targeting ABPs KY371 (*N*-alkynyl-cyclophellitol aziridine) and JJB111 (the biotin equivalent of KY371)[49,50], to *P. chrysosporium* cultures grown in minimal medium with beech wood chips as sole carbon and energy source (Fig. 1c). FP-alkyne is a reporter-tagged derivative of the well-known fluorophosphonate serine hydrolase inhibitors and thus specifically binds all SHs, i.e., serine proteases and metabolic SHs, without specificity within this enzyme class[51]. By contrast, JJB111 is an aziridine analog of cyclophellitol, a natural product inhibitor of GHs. JJB111 structurally mimics β-glucopyranoside moieties and, therefore, preferentially reacts with retaining β-glucosidases[49]; in addition, it however also labels a variety of β-exoglycosidases[52,53]. In hemicellulose degradation, the acetyl-xylan esterases, among them members of the SH family, cleave as one of the first steps in the overall degradation pathway acetyl groups from the carbohydrate/polysaccharide backbone[54]. The GHs, in turn, are responsible for cellulose, pectin, and xylan hydrolysis. In addition to extracellular soluble enzymes found in the culture supernatant, we also applied our ABPP approach to substrate-bound enzymes isolated from wood chips. Our results thus demonstrate that ABPP allows a straightforward and technically simple targeted identification of active biocatalysts, including enzymes with previously unannotated gene sequences (composed of domains of unknown function (DUF)), directly from fungal/microbial cultures grown on complex lignocellulosic substrates.

## Results

**ABPP analysis of the *Phanerochaete chrysosporium* supernatant preparation.** Previous studies on the identification of *P. chrysosporium* lignocellulose-degrading enzymes have been performed *via* 'classical' full proteome analyses of culture supernatants[55]. To demonstrate the potential of ABPP in the rapid identification of lignocellulolytic enzymes, we grew *P. chrysosporium* cultures (DSM 1566) for 5 days at 37 °C in beech wood chips-containing minimal medium (Fig. 2a). The formation of fungal hyphae as well as macroscopic degradation of growth substrates in a submerged liquid culture confirmed efficient fungal cell growth under these conditions. The culture supernatant was filtrated, lyophilized, re-dissolved in buffer, and then labeled with 2 µM of the two ABPs FP-alkyne or JJB111, respectively. For competitive ABPP experiments, pretreatment with either 50 µM paraoxon in the case of FP-alkyne or 20 µM KY358 in the case of JJB111 labeling was used. After affinity enrichment, target identification was achieved by on-bead tryptic digestion and LC-MS/MS analysis. Each identified protein was quantified using spectral intensity-based relative quantification; as a reference, DMSO- or corresponding competitor-treated samples were used. Only protein groups with a $\log_2$ fold change (FC) of ≥2 were kept for further analysis in all ABPP-based labeling experiments.

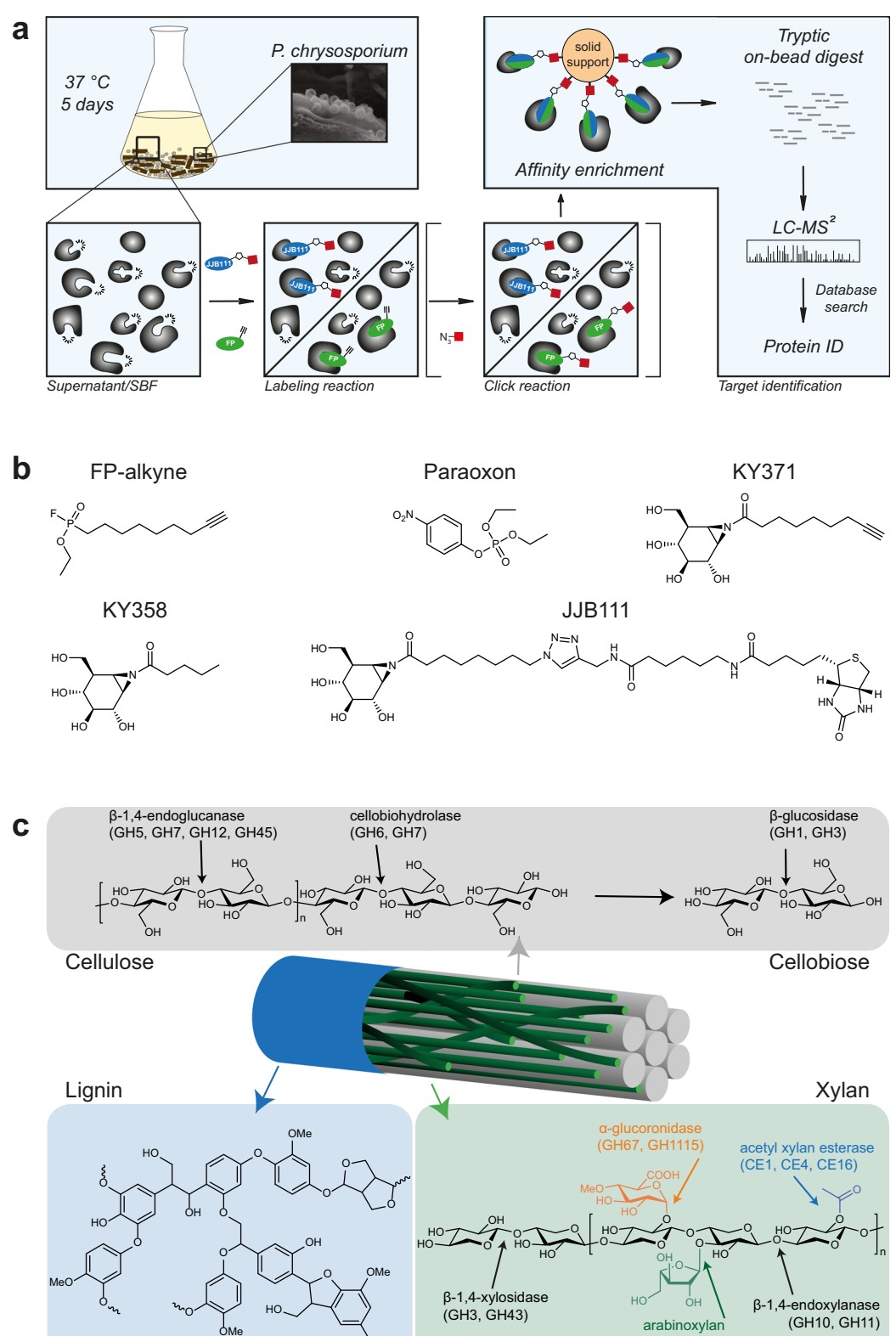

For the FP-alkyne treatment, this approach led to the identification of two SHs with the Joint Genome Institute (JGI) protein ID Phchr2|126075 (predicted carbohydrate esterase family 1 (CE1) with a CBM1 domain, MW of 35.6 kDa) and Phchr2|2912243 (predicted carbohydrate esterase family 15 (CE15), MW of 44.3 kDa) (Table 1 and green-labeled proteins in Fig. 2b; see Supplementary Data 1 for the complete list of identified proteins). Phchr2|126075 displays high sequence similarity to a previously studied acetyl-xylan esterase of *P. chrysosporium*, and its labeling was competed by pretreatment with the esterase inhibitor paraoxon[56]. In contrast, Phchr2|2912243 sequence analysis suggests this enzyme to be a CE15 family member and, therefore likely to be a 4-*O*-methyl-glucuronoyl methylesterase.

**Fig. 1 Activity-based protein profiling (ABPP) for identifying lignocellulose-degrading biocatalysts from *P. chrysosporium*. a** Overview on the ABPP workflow for identifying lignocellulose-degrading enzymes from *P. chrysosporium* suspension cultures grown on minimal medium with beech wood chips. An ABP is added to either the filtrate of a *P. chrysosporium* beech wood culture (denoted as supernatant) or the dodecylmaltoside-solubilized substrate-bound fraction (denoted as SBF) after lyophilization. The pretreatment is followed by a standard ABPP workflow, consisting of click-attachment of a biotin-residue for affinity enrichment (in case of a two-step ABP), affinity enrichment of labeled enzymes, trypsin digest, and subsequent MS-based protein identification. The use of enzyme class-specific ABPs, therefore, results in targeted identification of active biocatalysts and thus functional enzyme screening and also enables the sequence-independent identification of novel biocatalysts without similarity to known homologous. The inlet image shows how *P. chrysosporium* binds to the solid wood surface during lignocellulose degradation. **b** Chemical structures of the ABPs and competitors used in this study. These are FP-alkyne (the "classical" serine hydrolase ABP) and JJB111 (GH ABP), as well as the FP competitor paraoxon and the JJB111 competitors KY371 and KY358. **c** Lignocellulose is a complex and recalcitrant polymer built up from cellulose, xylan (hemicellulose), and lignin. Its degradation requires the synergistic action of various different enzymes.

The application of JJB111 enabled the identification of twelve GHs (Table 1 and blue-labeled proteins in Fig. 2c; see Supplementary Data 2 for the complete list of identified proteins). Pretreatment with KY358 competed labeling of six of them belonging to the GH3, GH5, GH16, and GH74 families. Interestingly, in contrast to the identified GHs known to be involved in cellulose (GH3 and GH5), xylan (GH3), or xyloglucan (GH74) degradation, our analysis also revealed significant enrichment ($\log_2$ fold change: 5.39) of the protein Phchr2|3002168, which is annotated as a glutaminase in the JGI MycoCosm genome database[45] (Table 1 and red-labeled protein in Fig. 2c). The labeling of Phchr2|3002168 was competed by pre-incubation with KY358 and domain analysis by PFAM[57] and InterProScan[58] of its protein sequence revealed the presence of four DUF domains (DUF4964, DUF5127, DUF4965, and DUF1793) along with a secretion signal (Supplementary Fig. 1a). Structural homology analysis by HHpred[59] however predicted a β-glucosidase of the GH116 family from *Thermoanaerobacterium xylolyticum* (pdb code 5O0S[60], e-value of $7.2e^{-34}$, 14% sequence identity) and a GH52 xylosidase from *Geobacillus thermoglucosidasius* (pdb code 4C1O[61], e-value of $7.1e^{-31}$, 10% sequence identity) as homologous proteins (Supplementary Fig. 1b). Additionally, analysis with InterProScan suggested the presence of a six-hairpin glycosidase domain which is characteristic for GH15, GH65, GH92 and GH116 family members. The Alphafold[62]-predicted 3D structure of Phchr2|3002168 showed a high secondary structure overlap to the GH52 β-xylosidase 4C1P (($\alpha/\alpha)_6$ barrel) of 70% even despite its low sequence similarity of 12% (Supplementary Fig. 1c, d). Overall, these results indicate that Phchr2|3002168 might be a GH of a so far uncharacterized GH family.

**ABPP analysis of the *P. chrysosporium* substrate-bound fraction (SBF).** Our study so far demonstrates that ABPP can be used to detect active SH and GH enzymes in the fungal culture supernatant. However, lignocellulose degradation by *P. chrysosporium* involves a variety of enzymes, which partially attach to the respective carbohydrate-based substrate[63]. Naturally, these substrate-bound enzymes are of particular interest for biotechnological applications as they are directly involved in lignocellulose degradation. During sample preparation, the lignocellulose substrate was filtered off to obtain a homogenous starting material for labeling. Indeed, such a discarding step is frequently performed in diverse functional screening approaches and a technically simple approach for targeted analysis of substrate-attached biocatalysts would be highly desirable.

To investigate whether such enzymes can be detected by a substrate-targeted ABPP approach, we again grew *P. chrysosporium* in the presence of beech wood chips. After removal of the culture supernatant and free fungal cells, the active substrate-bound enzymes were detached with 0.1% (w/v) of the MS-compatible detergent dodecyl-β-D-maltoside. The detached proteins were then lyophilized and the residue was resolved in buffer,

followed by the addition of the ABPs and the standard downstream MS sample preparation and analysis workflow (Fig. 3a).

The employment of FP-alkyne led to the identification of 53 substrate-bound proteins, of which seventeen were enriched with a $\log_2$-fold change $\geq 2$. Their functional annotation revealed the presence of four CEs from the CE1 and CE15 families (Table 2 and green-labeled proteins in Fig. 3b; see Supplementary Data 3 for the complete list of identified proteins). Notably, only the labeling of the CE1 family proteins Phchr2|2983171 and Phchr2|126075 was successfully inhibited by pre-incubation with paraoxon. These proteins are of potential biotechnological interest due to their predicted potential as acetyl or feruloyl esterases. Moreover, five serine carboxypeptidases (S10) and four carboxylesterases were identified which may play a role in enzyme activation during lignocellulose degradation, for instance of cellobiose dehydrogenases[64]. The labeling of three of the enriched carboxylesterases was furthermore competed by paraoxon.

The analysis of the ABPP approach with JJB111 revealed seven GHs, which were enriched with a $\log_2$-FC $\geq 2$ (blue-labeled proteins in Fig. 3c, see Supplementary Data 4 for the complete list of identified proteins). Of these, four were competed by pre-incubation with KY371. The proteins Phchr2|3002242, Phchr2|2945552, and Phchr2|3003144 are predicted members of the GH3 family, whereas protein Phchr2|2915237 belongs to GH5 subfamily 9; both GH families are known to catalyze cellulose or xylan degradation. By contrast, the labeling of the proteins Phchr2|3004009 (GH17), Phchr2|2895579 (GH5), and Phchr2|3038646 (GH25) was not competed by KY371. Of note, three of the identified proteins (i.e., Phchr2|291537, Phchr2|126075, and Phchr2|2912243) were also identified in the ABPP analysis of the supernatant.

Overall, these experiments demonstrate that ABPP is not only a technically simple targeted approach to identify promising lignocellulose biocatalysts but also enables to focus the analysis on relevant, active enzyme subfractions, e.g., those bound to an insoluble substrate.

**Biochemical validation and characterization of selected identified enzymes.** So far, our ABPP approach identified several potential lignocellulose-degrading biocatalysts. Their subsequent functional annotation was achieved by sequence homology analysis. The ABPP approach may, however, in principle, also enable the identification of enzymes of a new enzyme family, as enzyme identification is based on the ABP enzyme reactivity and not sequence homology. To demonstrate that our ABPP approach indeed resulted in the identification of lignocellulose-degrading biocatalysts, we selected three of the ABPP-identified enzymes, Phchr2|126075, Phchr2|2915237, and Phchr2|3002168, for further expression and subsequent biochemical characterization (Supplementary Fig. 2).

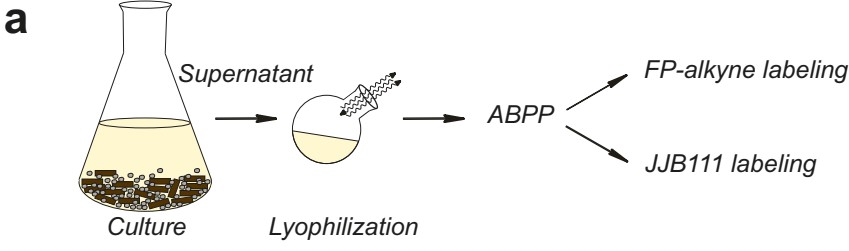

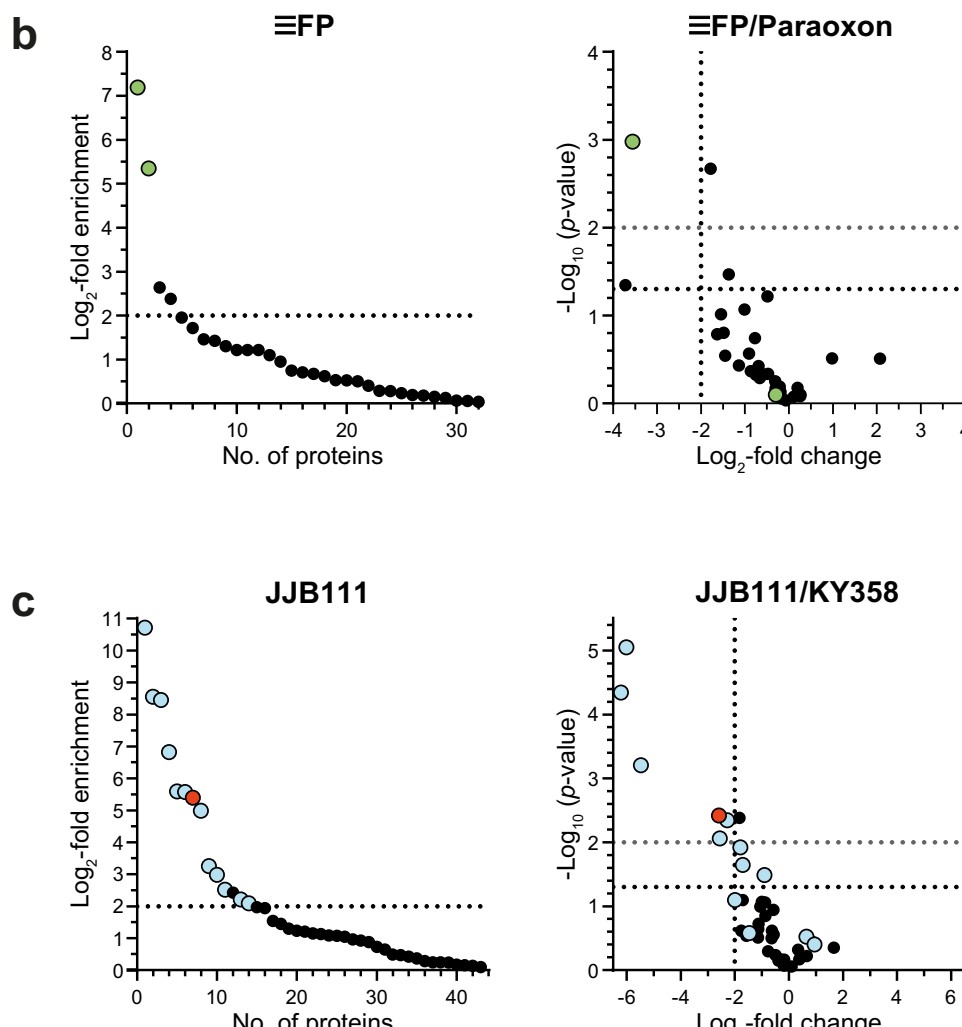

**Fig. 2 ABPP of *P. chrysosporium* supernatants. a** Workflow of the analysis. *P. chrysosporium* suspension cultures were grown for 5 days on a minimal medium supplemented with beech wood chips. The solid material was filtered off, the filtrate lyophilized, and the residue was subjected to ABPP with the corresponding probes. **b** ABPP of SHs without (left panel, log$_2$-fold change ≥2 compared to DMSO is indicated) or after pretreatment with 50 μM paraoxon (competition experiment, right panel) with 2 μM FP-alkyne after click chemistry ($n = 4$ biologically independent samples). Green dots indicate SHs. **c** ABPP of GHs without (left panel, log$_2$-fold change ≥2 compared to DMSO is indicated) or after pretreatment with 20 μM KY358 (competition experiment, right panel) with 2 μM JJB111 ($n = 4$ biologically independent samples). Blue dots indicate GHs, while the red dot represents the DUF protein with four domains of unknown function Phchr2|3002168 with potential GH activity.

*Acetyl-xylan esterase activity of Phchr2|126075.* The putative acetyl-xylan esterase Phchr2|126075 (338 amino acids; 35.5 kDa) was identified with FP-alkyne in the *P. chrysosporium* supernatant (log$_2$-fold enrichment of 7.18) and in the SBF (log$_2$-fold enrichment of 2.37). Phchr2|126075 belongs to the CE1 family and contains a fungal CBM1 motif for carbohydrate binding as well as

a secretion signal. The esterase domain encompasses the residues 80-288. Of note, a homologous acetyl-xylan esterase (Phchr2|129015, e-value of 0.0, 89% sequence identity) from the same species was previously characterized[56]. For characterization, *phchr2|126075* was cloned into the pKLAC2 vector and over-expressed in *Kluyveromyces lactis*. As expected from the sequence

**Table 1 Overview of the ABPP-identified enzymes from profiling the supernatant of *P. chrysosporium* suspension cultures grown on minimal medium supplemented with beech wood chips.**

| Protein ID | Annotation | Used ABP | Log$_2$ fold change | MW [kDa] | Predicted function |
|---|---|---|---|---|---|
| Phchr2\|126075 | CE1-CBM1 | FP-alkyne | 7.19 | 35.59 | acetyl-xylan esterase |
| Phchr2\|2912243 | CE15 | FP-alkyne | 5.35 | 44.29 | 4-O-methyl-glucuronoyl methylesterase |
| Phchr2\|2915237 | GH5 subfamily 9 | JJB111 | 10.71 | 46.55 | 1,4-β-endoglucanase |
| Phchr2\|3024052 | GH3 | JJB111 | 8.55 | 86.85 | 1,4-β-xylosidase |
| Q66NB7; Phchr2\|2985730 | GH5 subfamily 6 | JJB111 | 8.45 | 40.41 | 1,4-β-glucanase |
| Q9URP5;O74203 | CBM1-GH3 | JJB111 | 6.82 | 85.56 | β-glucosidase |
| Phchr2\|2895579 | GH5 subfamily 9 | JJB111 | 5.59 | 73.50 | 1,3-β-glucosidase |
| Phchr2\|2990154 | GH30 subfamily 3 | JJB111 | 5.58 | 65.13 | 1,6-β-endoglucanase |
| Phchr2\|3002168 | DUF | JJB111 | 5.39 | 74.43 | DUF1793, DUF4964, DUF4965, DUF5127 |
| Phchr2\|3004260 | GH79 | JJB111 | 4.99 | 48.99 | β-glucuronidase |
| Phchr2\|2981757 | GH5 subfamily 5 | JJB111 | 3.26 | 42.05 | 1,4-β-endoglucanase |
| Phchr2\|2927412 | GH74 | JJB111 | 2.99 | 79.15 | exo-1,3-β-xyloglucanase/reducing end-specific cellobiohydrolase |
| Phchr2\|2916357 | GH28 | JJB111 | 2.52 | 53.53 | galacturonase |
| W5ZNX7;C6H06; Phchr2\|123909 | GH16 | JJB111 | 2.21 | 33.57 | endo-1,3-β-glucanase |
| Phchr2\|3006243 | GH18, CBM5 | JJB111 | 2.09 | 54.31 | endo-1,3-β-glucanase |

Two SHs, 12 GHs, and one protein (DUF) with four DUFs with similarities to GHs were elucidated from FP-alkyne and JJB111 labeling, respectively.

homology, Phchr2|126075 displayed esterase activity against *p*NP-acetate and subsequent assays revealed the highest enzyme activity at a pH of 8 and 40 °C, respectively (Supplementary Fig. 3). Further kinetic characterizations then revealed a $V_{max}$ of 41.7 U mg$^{-1}$ protein and $K_M$ of 0.67 mM for *p*NP-acetate hydrolysis (Fig. 4a).

*β-glucanase activity of Phchr2|2915237.* Phchr2|2915237 (422 amino acids; 46.5 kDa) was identified and enriched with JJB111 in both the soluble supernatant (log$_2$ fold change: 10.71) and the SBF (log$_2$ fold change: 4.11). An analysis with InterProScan indicates the presence of a GH5 cellulase domain encompassing residues 67-330. Phchr2|2915237 also contains an extracellular secretion signal but no transmembrane domains. A HHpred[59] analysis predicts either β-1-3 glucanases[65] (e-value: 9.7e$^{-35}$; 44% sequence identity:) or β-1-4-xyloglucanases[66] (e-value: 8.8e$^{-24}$, 18% sequence identity) as the closest structural homologs. Additionally, homologs of Phchr2|2915237 are present in different wood-degrading fungal species, such as *Trametes* or *Pleurotus* as identified via BLASTP[67]. The closest characterized homolog (e-value: 9e$^{-108}$, 45% sequence identity) of Phchr2|2915237 belongs to the yeast *Candida albicans* and plays a role in cell wall metabolism and restructuring[68].

We heterologously expressed Phchr2|2915237 in *Aspergillus oryzae* and studied its substrate specificity using a variety of chromogenic *p*-nitrophenol-sugar conjugates as well as different polysaccharides after purification. Phchr2|2915237 showed high GH activity if *p*NP-Glc and *p*NP-Xyl were used as substrates, while with *p*NP-Ara only residual and with *p*NP-Man, *p*NP-GlcNAc, and *p*NP-Gal no hydrolytic activity was observed (Fig. 4b). For *p*NP-Glc hydrolysis, a pH and temperature optimum of 5 and 60 °C, respectively, was elucidated (Supplementary Fig. 3). A 3, 5-dinitrosalicylic acid (DNSA) assay revealed that Phchr2|2915237 was also able to degrade both lichenan and beech wood xylan, while carboxymethyl cellulose (CMC), galactomannan, xyloglucan, and curdlan were no suitable substrates (Fig. 4b). More detailed kinetic characterizations revealed a $V_{max}$ of 999 U mg$^{-1}$ protein and a $K_M$ of 1.82 mM as well as a $V_{max}$ of 612 U mg$^{-1}$ protein and a $K_M$ of 6.98 mM for *p*NP-glucopyranoside and *p*NP-xylopyranoside, respectively. For polysaccharide degradation, a $V_{max}$ value of 107 U mg$^{-1}$ protein

with a $K_M$ value of 5.5 mg mL$^{-1}$ as well as a $V_{max}$ value of 71.6 U mg$^{-1}$ protein and a $K_M$ value of 13.8 mg mL$^{-1}$ was determined with lichenan and beech wood xylan, respectively, showing that Phchr2|2915237 is also able to cleave natural, more complex sugar polymers (Fig. 4c). To determine if Phchr2|2915237 functions as an exo- or endo-glucanase/xylanase, we analyzed the hydrolysis products of xylan and lichenan generated by Phchr2|2915237 via thin-layer chromatography. No monosaccharides were cleaved from the glucan chains; we instead observed the formation of polysaccharide degradation products with unknown length (Supplementary Fig. 4). In addition, a coupled assay using either a glucose or xylose dehydrogenase also showed no formation of either glucose or xylose by Phchr2|2915237 during hydrolysis of lichenan and xylan, respectively.

*Activity of Phchr2|3002168 DUF family protein.* Phchr2|3002168 (691 amino acids; 74.4 kDa) is an uncharacterized protein that was enriched by JJB111; its JJB111 labeling was competed by pretreatment with KY371. The four-domain protein Phchr2|3002168 is annotated as a glutaminase, but our ABPP approach, as well as our previously described additional sequence analyses suggested a possible GH activity. Moreover, a full proteome analysis of different fungal species including *P. chrysosporium* revealed that some of them secreted Phchr2|3002168-homologous proteins in the presence of lignocellulose[69].

We, therefore, tried to characterize this protein via biochemical assays. However, all our attempts to express and purify Phchr2|3002168 in *E. coli* or *A. oryzae* failed, emphasizing again the persisting difficulties of an expression- vs. an ABPP-based functional biocatalyst screening of fungal proteins. To confirm the activity of Phchr2|3002168 as a GH, we, therefore, searched for close homologs in other organisms and found WP_074995790 from *Streptomyces misionensis* (e-value: 0.0; 42% sequence identity) as a promising candidate. Please note that although WP_074995790 displays the same four domain structure as Phchr2|3002168, the InterPro protein annotation just lists the DUF5127 domain.

We, therefore, overexpressed WP_074995790 (752 amino acids; 80.5 kDa) and, although this enzyme was difficult to handle due to strong aggregation tendency in biochemical assays, we were able to perform some enzyme assays with this

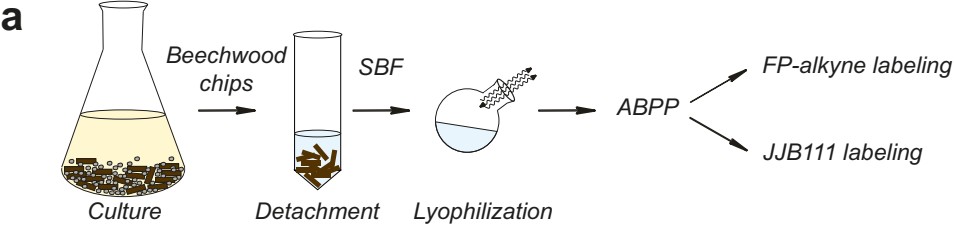

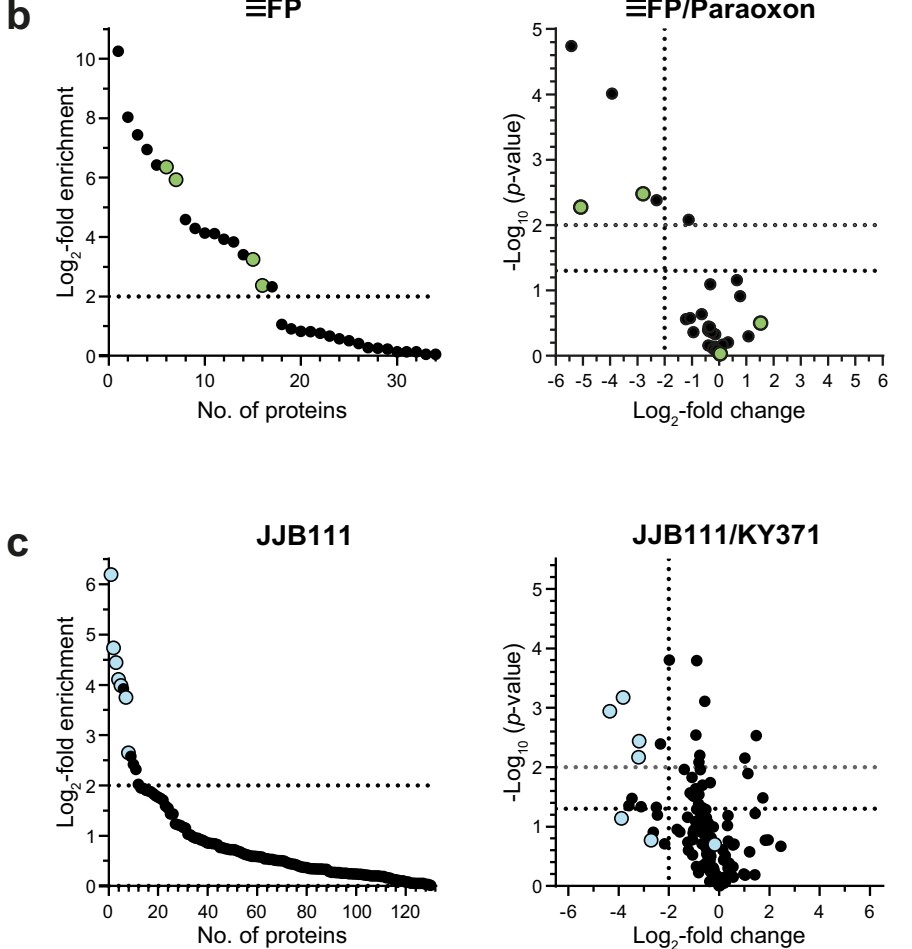

**Fig. 3 ABPP of *P. chrysosporium* substrate-bound fraction (SBF). a** Workflow of the SBF analysis. *P. chrysosporium* suspension cultures were grown for 5 days on a minimal medium supplemented with beech wood chips. The beech wood chips were isolated and substrate-bound proteins were isolated via 0.1% (w/v) dodecylmaltoside treatment. The obtained protein solution was lyophilized and the residue was subjected to ABPP with the corresponding probes. **b** ABPP of SHs without (left panel, log$_2$-fold change ≥2 compared to DMSO is indicated) or after pretreatment with 50 µM paraoxon (competition experiment, right panel) and 2 µM FP-alkyne after click chemistry ($n = 4$ biologically independent samples). Green dots indicate annotated CEs. **c** ABPP of GHs without (left panel, log$_2$-fold change ≥2 compared to DMSO is indicated) or after pretreatment with 20 µM KY371 (competition experiment, right panel) with 2 µM JJB111 ($n = 3$ biologically independent samples). Blue dots indicate annotated GHs.

preparation. Due to its original assignment as a glutaminase, we started with corresponding glutaminase enzyme assays but were unable to detect any conversion of a glutamine substrate. We next screened for GH activity by testing the same set of *p*NP-based sugar substrates used before with Phchr2|2915237. Satisfactorily, we were able to detect a weak β-galactosidase activity with a total specific activity of 0.85 U mg$^{-1}$ protein. All other tested *p*NP-substrates were, however, not hydrolyzed (Fig. 4b). Altogether, these biochemical assays, therefore, demonstrated a GH activity of WP_074995790, although the observed overall weak β-

galactosidase activity predicts that other carbohydrate structures, e.g., more complex polymeric carbohydrates, may represent better substrates.

## Discussion

White rot fungi exhibit excellent decomposition abilities and are responsible for the degradation of lignin in plant biomass; they have therefore attracted considerable interest as resources for identifying biotechnologically-relevant enzymes[56,70]. Among them, the species *P. chrysosporium* seems to be particularly well-suited as

**Table 2 Overview of the ABPP-identified enzymes from profiling the substrate-bound fraction (SBF) of *P. chrysosporium* suspension cultures grown on minimal medium supplemented with beech wood chips.**

| Protein ID | Annotation | Used ABP | Log₂ fold change | MW [kDa] | Predicted function |
|---|---|---|---|---|---|
| Phchr2\|3002242 | GH3 | JJB111 | 6.20 | 92.86 | β-glucosidase |
| Phchr2\|3004009 | GH17 | JJB111 | 4.73 | 33.78 | 1,3-β- glucanosyltransferase |
| Phchr2\|2895579 | GH5 | JJB111 | 4.44 | 73.50 | 1,3-β-glucosidase |
| Phchr2\|2915237[a] | GH5, subfamily 9 | JJB111 | 4.11 | 46.55 | 1,4-β- endoglucanase |
| Phchr2\|2945552 | GH3-WSC | JJB111 | 3.99 | 98.95 | β-glucosidase |
| Phchr2\|3038646 | GH25 | JJB111 | 3.75 | 24.47 | muramidase |
| Phchr2\|3003144 | GH3 | JJB111 | 2.65 | 92.83 | β-glucosidase |
| Phchr2\|2912243[a] | CE15 | FP-alkyne | 6.35 | 44.29 | glucuronoyl methylesterase |
| Phchr2\|2983171 | CE1, CBM1 | FP-alkyne | 5.93 | 31.85 | acetyl-xylan esterase |
| Phchr2\|2918304 | CE15, CBM1 | FP-alkyne | 3.24 | 47.47 | glucuronoyl methylesterase |
| Phchr2\|126075[a] | CE1, CBM1 | FP-alkyne | 2.37 | 35.59 | acetyl-xylan esterase |

[a]Enzymes that were also labeled in the supernatant (Table 1).
Four SHs and seven GHs were elucidated from FP-alkyne and JJB111 labeling, respectively.

a starting point for biocatalyst discovery as its enzymes are often thermostable due to its rather high growth temperature optimum of 40 °C[71]. Accordingly, multiple proteomic studies have investigated its secretome during growth on lignocellulose substrates with the aim of identifying lignocellulose-degrading biocatalysts, although most of the identified biocatalysts have never been validated biochemically[46,55,69,72].

Here we described an alternative approach for identifying biotechnologically-relevant enzymes via ABPP with enzyme class-specific ABPs that allows to focus the analysis on only active enzymes with relevance to the overall biological process. This approach can be employed to rapidly analyze extracellular soluble (supernatant) biocatalysts. More importantly, however, it can also be used, as described here for the first time, to identify substrate-bound biocatalysts, e.g., in our case, enzymes attached to solid lignocellulose in the form of beech wood chips. The ABPP approach represents a preselection step for the most promising biocatalysts. Its application to soluble enzymes or enzymes directly bound to substrates thereby allows a technically straightforward functional screening that we anticipate may find wider application in targeted biocatalyst discovery. The use of further ABPs, e.g., GH-directed probes with different α/β-specificity or different sugar selectivity[50], will thereby enable to elucidate additional enzymes for lignocellulose degradation.

To demonstrate the applicability of this approach, we biochemically validated three of the identified ABPP 'hits'. We selected one SH target of the FP-alkyne as well as one GH target of the JJB111 ABPP labeling approach. In addition, we chose to characterize a bacterial homolog of one target enzyme, Phchr2\|3002168, which has not been assigned as a carbohydrate-active enzyme (CAZyme) based on sequence analyses. The FP-alkyne identified CE1 family protein Phchr2\|126075 was homologous to a characterized acetyl-xylan esterase[73], and we were able to confirm a potent hydrolytic activity against *p*NP-acetate. This suggests a potential biotechnological application of this enzyme in the first steps of hemicellulose degradation, which is the cleavage of acetyl groups on lignocellulose. Phchr2\|2915237 was identified as a promiscuous and highly catalytically active polysaccharide-cleaving β-endoglucanase that shows activity against both xylan and lichenan as well as against *p*NP-Glc and *p*NP-Xyl and releases neither glucose nor xylose from lichenan or xylan chains, respectively. It therefore most likely contributes to the breakdown of lignocellulose in *P. chrysosporium*. Phchr2\|2915237 contains a secretion signal for extracellular transport and is predicted to belong to the GH5 subfamily 9. So far, only a few enzymes from this subfamily are known, most of which contain exo-β-1,3- or

exo-β-1,4-glucanase activity in addition to an endo-1,6-glucanase activity in some family members. A total of 17 family members have been characterized, all of them either belonging to different yeast or *Aspergillus* species, while no homolog has been described in any basidiomycete so far. Characterized homologs of Phchr2\|2915237 have been found to play key roles in morphogenetic processes during development and differentiation, for example, in *Candida albicans* where exo-β-1,3-glucanases partially hydrolyze cell wall areas, enabling the insertion of new cell wall material, and can additionally also cleave 1,4 and 1,6 glycosidic bonds[68]. In *S. pombe*, a GH5 subfamily 9 protein was able to hydrolyze both β-1,3 and β-1,6 glycosidic bonds[74,75]. However, different homologs have also been shown to function as antifungal enzymes or to be involved in plant cell wall degradation[76,77]. Interestingly, Phchr2\|2915237 also shows some similarity to endo-1-4-glucanases that have been shown to catalyze the cleavage of different xylo/gluco-oligosaccharides[66]. Regarding the cleavage of fungal cell walls in *P. chrysosporium*, so far, mostly GH16 and GH55 enzymes have been attributed to be involved in cell wall morphogenesis and nutrient recycling[78]. Although we do not know the exact in vivo functions of Phchr2\|2915237, its export seems to be induced when *P. chrysosporium* is grown on lignocellulose and its dual endo-glucanase/xylanase activity would allow the enzyme to take part in the degradation of plant cell wall material. The overall high enzymatic activity and the broad substrate specificity of Phchr2\|2915237, in conjunction with its compatibility with heterologous expression in an industrial-relevant production organism such as *A. oryzae* turns this enzyme into a promising biocatalyst for efficient carbohydrate degradation. Finally, we were able to show the potential of ABPP to annotate also proteins of unknown or misassigned function. Phchr2\|3002168 was identified by labeling with the GH probe JJB111. As we, however, failed to directly express this protein, we instead characterized the highly homologous protein WP_074995790 from *S. misionensis*, a bacterium also known to degrade cellulose in the form of sugarcane bagasse[79]. We could confirm a β-galactosidase activity even though the total specific activity was low. This indicates that WP_074995790 might indeed possess GH activity but that we were not able to identify the native substrates for these novel enzymes so far. However, based on our labeling approach, sequence analysis, and enzyme assays, we suggest that proteins containing DUF4964, DUF5127, DUF4965, and DUF1793 domains, such as Phchr2\|3002168 or WP_074995790 may function as glycoside hydrolases.

However, it should be noted that the reported ABPP approach in this study has also some limitations. The identification of

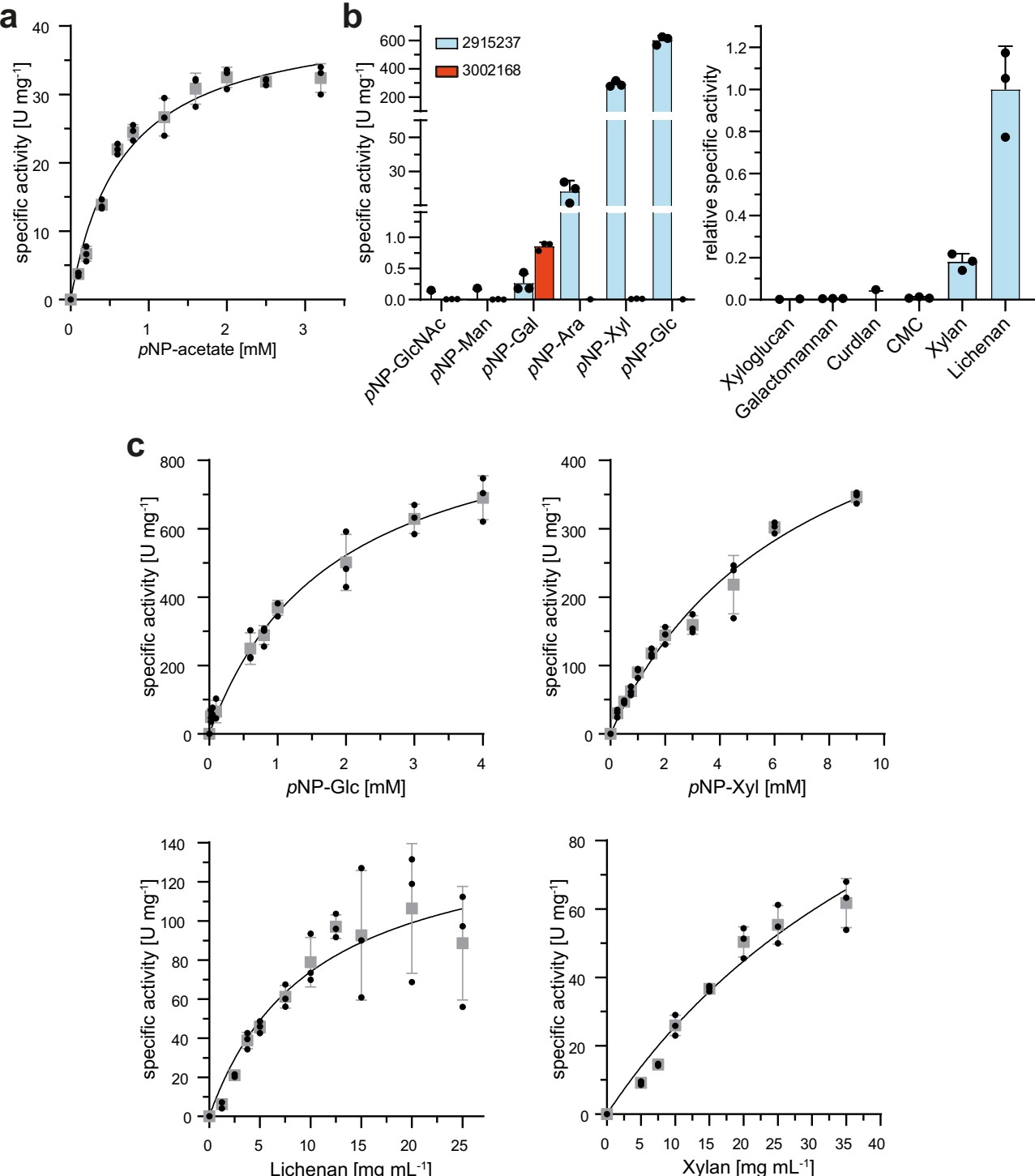

**Fig. 4 Biochemical characterization of selected identified enzymes. a** Phchr2|126075 was heterologously produced in *K. lactis* and activity was confirmed against *p*NP-acetate by following the release of *p*NP with a $V_{max}$ of 41.7 U mg$^{-1}$ protein and a $K_M$ of 0.67 mM. **b** Substrate specificity of Phchr2|2915237 and WP_074995790 from *S. misionensis*. The activity with different *p*-nitrophenol-based substrates was determined by measuring the release of *p*NP. The activity on complex polysaccharides was determined via the DNSA assay that quantifies the formation of reducing ends upon polysaccharide cleavage. Phchr2|2915237 displayed the highest activities with *p*NP-Glc, *p*NP-Xyl, lichenan, and beech wood xylan, while WP_074995790, a close homolog of Phchr2|3002168 showed activity only with *p*NP-Gal as a substrate with a specific activity of 0.85 U mg$^{-1}$ protein. **c** Kinetic characterization of Phchr2|2915237 using *p*NP-Glc, *p*NP-Xyl, lichenan, and beech wood xylan as substrates. All activity measurements were performed in triplicate ($n = 3$), mean values are shown and the error bars indicate the standard deviation (SD).

potential ABPP target proteins is influenced by the cell culturing conditions as the composition of proteins secreted by *P. chrysosporium* is dependent on culture growth time and the source and treatment of lignocellulosic biomass. Since both factors were fixed in this study, the scope of potential targets is limited to the proteins produced and secreted under these conditions. In addition, the usage of additional ABPs, preferentially with different target specificity, may allow the identification of further enzymes. ABPP labeling may also be lowered by the presence or enzyme-mediated production of competing metabolites. Finally, most ABBP methods currently require subsequent bioinformatics analysis and the verification of target hits via expression and purification due to a small proportion of unspecific labeling during ABPP profiling.

In conclusion, our ABPP approach may help to overcome a persisting challenge in biocatalyst discovery: the difficulty to link data from a functional screen to sequence information. The ABPP approach shortcuts this and allows to narrow the analysis to active enzymes targeted by the enzyme-selective ABP probe. As more and more ABPs become available, this may allow to identify biocatalyst ensembles even for the degradation of complex substrates solely by assembling together identified enzymes from different ABPP screening campaigns.

## Methods

**Chemicals**. Chemicals for cultivation of *Escherichia coli* and *P. chrysosporium* DSM 1556, including yeast extract, malt extract, soytone, lysogeny broth, TRIS, MES, and salts for minimal media were obtained from Carl Roth (Germany). Beech wood chips for growth were obtained from J. Rettenmaier Söhne GmbH & Co. KG. (Germany). Carboxymethyl cellulose (CMC), lichenan, mannan, xyloglucan, and glucomannan were purchased from Sigma Aldrich (USA), and beech wood xylan was purchased from Carl Roth (Germany). *para*-nitrophenol (*p*NP), *para*-nitro-phenyl-β-D-galactopyranoside (*p*NP-Gal), *para*-nitrophenyl-acetate (*p*NP-acetate), *para*-nitrophenyl-β-D-glucopyranoside (*p*NP-Glc), *para*-nitrophenyl-β-D-xylopyr-anoside (*p*NP-Xyl), *para*-nitrophenyl-β-D-mannose (*p*NP-Man), *para*-nitrophenyl-β-D-arabinofuranoside (*p*NP-Ara), and *para*-nitrophenyl-N-acetyl-β-D-glucosa-mine (*p*NP-GlcNAc) were purchased from Megazyme (Ireland), *n*-dodecyl β-D-maltoside (DDM) was obtained from Thermo Scientific (USA) and bovine serum albumin (BSA) from VWR Chemicals (USA). The sources of supply of more methodology-specific reagents are reported in the corresponding procedure section.

**Cell growth and storage**. *P. chrysosporium* DSM 1556 was obtained from the DSMZ (Germany). For long-term storage *P. chrysosporium* DSM 1556 was grown on MYP-agar plates (6 g L$^{-1}$ malt extract; 1 g L$^{-1}$ peptone from soy, 0.5 g L$^{-1}$ yeast extract) for 2 days at 37 °C. Afterward, cells were scraped off, resuspended in 100 µL aliquots of sterile 50% (v/v) glycerol and stored at −80 °C as a glycerol stock. For the growth of solid cultures, 1.5% (w/v) MYP-Agar was inoculated with 20 µL of *P. chrysosporium* DSM 1556 glycerol stock and grown for 2 days at 37 °C until the whole agar plate was covered with fungal hyphae. Afterward minimal medium containing 2.5 g L$^{-1}$ K$_2$HPO$_4$, 0.02 g L$^{-1}$ KH$_2$PO$_4$, 0.1 g L$^{-1}$ NaCl, 0.02 g L$^{-1}$ CaCl$_2$, 0.1 g L$^{-1}$ (NH$_4$)$_2$SO$_4$, 0.02 g L$^{-1}$ MgSO$_4$, 0.001 g L$^{-1}$ FeSO$_4$, and 40 g L$^{-1}$ beech wood chips at pH 5 supplemented with 100 µg mL$^{-1}$ of chloramphenicol to inhibit bacterial growth were inoculated with plate grown *P. chrysosporium* DSM 1556. Suspension cultures were incubated for 5 days at 37 °C under constant shaking (180 rpm). Cell growth was tracked by following the formation of fungal hyphae as well as by the macroscopic degradation of the growth substrate. *E. coli* Rosetta DE3 was either grown in standard LB-medium (10 g L$^{-1}$ tryptone, 10 g L$^{-1}$ NaCl, 5 g L$^{-1}$ yeast extract) in precultures or in TB-medium (22 g L$^{-1}$ yeast extract, 12 g L$^{-1}$ tryptone, 4 mL L$^{-1}$ glycerol, 0.072 M K$_2$HPO$_4$, 0.017 M KH$_2$PO$_4$, pH 7.2) for heterologous overexpression of enzymes. *Kluyveromyces lactis* GG799 was grown in supplied media of the NEB *K. lactis* protein expression kit (New England Biolabs, USA) or in YPGlu media (10 g L$^{-1}$ yeast extract, 20 g L$^{-1}$ peptone, 2 % glucose, pH 7) for heterologous overexpression.

**Production of supernatant and SBF**. After 5 days, the supernatant was removed and sterilized by filtration through a 0.2 µm filter (Filtropur S 0.2; Sarstedt, Germany). To obtain MS samples, 50 mL (labeling) culture supernatant was snap-frozen, lyophilized overnight, and subsequently stored at −20 °C until further analysis. For the isolation of substrate-bound proteins, *P. chrysosporium* DSM 1556 was grown on 40 g L$^{-1}$ beech wood chips in a minimal medium in 50 mL of submerged culture per replicate for a total of three or four biological replicates, respectively. After a growth time of 5 days, the beech wood chips were separated from the culture medium and free fungal cells by decantation and the remaining

wood chips were washed three times with 50 mL of buffer (50 mM TRIS, pH 8) by centrifugation (3000 × *g*, 10 min, 4 °C) to remove all unbound proteins and cells. The pelleted wood chips were then incubated in 5 mL of 50 mM TRIS pH 8 containing 0.1 % (w/v) of the MS-compatible detergent DDM at 37 °C for 30 min under constant shaking at 180 rpm to solubilize all substrate-bound proteins. Similar to the supernatant, 5 mL of the detachment solution was snap-frozen, lyophilized overnight, and subsequently stored at −20 °C until further analysis.

**Large-scale labeling for target identification**. All probes and competitors were dissolved in DMSO. To identify enzyme targets of FP-alkyne and JJB111, lyophilized proteins were resuspended in 2 mL (supernatant) or 100 µL (SBF) of either 50 mM Na$_2$PO$_4$ (pH 8.0) (FP-alkyne labeling) or 50 mM NaOAc (pH 5.0) (JJB111 labeling), respectively, and the protein concentration was determined with Roti®-Nanoquant (modified Bradford assay; Carl Roth, Germany). A total amount of 400 µg (supernatant) or 100 µg (substrate-bound fractions) protein was labeled with 2 µM of the indicated probe (1 h, 37 °C, vigorous shaking). For the competition of labeling with indicated ABPs, either 50 µM paraoxon (FP-alkyne labeling) or 20 µM KY358-acyl or KY371 (JJB111 labeling) was used as indicated (30 min, 37 °C, vigorous shaking). FP-alkyne labeled proteins were subsequently subjected to a click reaction with 10 µM TAMRA-biotin-N$_3$ (Jena Bioscience, Germany), 100 µM TBTA (Sigma Aldrich, USA), 2 mM TCEP (Sigma Aldrich, USA), and 1 mM CuSO$_4$ (Sigma Aldrich, USA; 1 h, room temperature, in the dark).

**Affinity enrichment and MS sample preparation**. Prior to affinity enrichment, a modified methanol-chloroform[80] precipitation was performed to clean up proteins. Briefly, protein solutions were incubated with four equivalents of methanol (−20 °C, overnight) before one equivalent of chloroform and three equivalents of MS-grade water (VWR Chemicals, USA) were added. The precipitated proteins were washed twice with methanol, dried on air, and then dissolved in a final volume of 8 mL 0.2 % (w/v) SDS in 1× PBS (155 mM NaCl, 3 mM Na$_2$HPO$_4$, 1.06 mM KH$_2$PO$_4$, pH 7.4) under gentle shaking (37 °C, 30 min). For enrichment of ABP-reacted proteins, the obtained protein solution was incubated with 100 µL of avidin bead slurry (Thermo Scientific, USA) while gently rotating (1 h, room temperature). Next, the beads were washed five times with 0.2 % (w/v) SDS (10 min, room temperature, gently rotating), followed by three washes with MS-grade H$_2$O (5 min, room temperature, vigorously shaking), and collected by centrifugation (400 × *g*, room temperature, 5 min). The washed beads were taken up in 100 µL of 0.8 M urea (GE Healthcare Life Sciences, USA) in 50 mM ammonium bicarbonate (ABC), the proteins were reduced with 5 mM dithiothreitol (DTT, Sigma Aldrich, USA) in 50 mM ABC (30 min, 37 °C, vigorous shaking), and subsequently alkylated by adding 10 mM iodoacetamide (IAM) in 50 mM ABC (30 min, 37 °C, in the dark). The alkylation reaction was quenched by adding DTT to a final concentration of 10 mM. For protein digestion, 1 µg trypsin (Thermo Fisher Scientific, USA) in 50 mM acetic acid was added (37 °C, 16 h, vigorous shaking). The beads were collected by centrifugation (5 min, room temperature, 650 × *g*) and the supernatant was recovered and mixed with formic acid (FA) to a final concentration of 0.5% (v/v). To wash the beads, 40 µL 1% (v/v) FA were added (5 min, room temperature, vigorous shaking) and the supernatant was combined with the recovered digestion mix. To remove the remaining beads from the peptide solution, the mix was centrifuged (5 min, room temperature, 100×*g*) through a homemade two-disc glass microfiber membrane (GE Healthcare, USA; pore size 1.2 µm, thickness 0.26 mm) StageTip. The cleared peptide solution was then desalted on homemade C18 StageTips as described (for the protocol used, see below).

**Sample clean-up for LC-MS/MS**. All peptide solutions after digestion and removal of solid matter were desalted using homemade C18 StageTips as described previously[81]. All centrifugation steps were performed in the range of 400–800 × *g* and for 1–3 min at room temperature. Briefly, the acidified tryptic digests were passed over two-disc StageTips and the immobilized peptides were washed twice with 0.5% (v/v) FA. The peptides were eluted from the StageTips by a two-step elution with 80% (v/v) acetonitrile containing 0.5% (v/v) FA. After elution from the StageTips, samples were dried using a vacuum concentrator (Eppendorf, Germany) and the peptides were resuspended in 15 µL 0.1% (v/v) FA. The thus prepared samples were directly used for LC-MS/MS experiments (see below for details).

**LC-MS/MS**. LC-MS/MS experiments were performed on an Orbitrap Fusion Lumos mass spectrometer (Thermo Fisher Scientific, USA) that was coupled to an EASY-nLC 1200 liquid chromatography (LC) system (Thermo Fisher Scientific, USA). The LC was operated in the one-column mode and the analytical column was a fused silica capillary (inner diameter 75 µm × 36–46 cm) with an integrated PicoFrit emitter (New Objective, USA) packed in-house with Reprosil-Pur 120 C18-AQ 1.9 µm (Dr. Maisch, Germany). The analytical column was encased by a column oven (Sonation, Germany) and attached to a nanospray flex ion source (Thermo Fisher Scientific, USA). The column oven temperature was adjusted to 50 °C during data acquisition. The LC was equipped with two mobile phases: solvent A (0.1% (v/v) FA, in water) and solvent B (0.1 % (v/v) FA, 20% (v/v) H$_2$O, in acetonitrile). All solvents were of UHPLC (ultra-high-performance liquid chromatography) grade (Honeywell, Germany). Peptides were directly loaded onto

the analytical column with a maximum flow rate that would not exceed the set pressure limit of 980 bar (usually around 0.5–0.8 µL min$^{-1}$). Peptide solutions were subsequently separated on the analytical column using different gradients (105 min length; for details, see Supplementary File Sample_Legend_and_LC-MS_Settings, Section "LC_Settings").

The mass spectrometer was operated using Xcalibur software v4.3.7.3.11. The mass spectrometer was set in the positive ion mode. Precursor ion scanning (MS$^1$) was performed in the Orbitrap analyzer (FTMS; Fourier Transform Mass Spectrometry with the internal lock mass option turned on (lock mass was 445.120025 $m/z$, polysiloxane))[82]. Dynamic exclusion was turned on (exclude after $n$ times = 1; Exclusion duration (s) = 120; mass tolerance = ±10 ppm). MS$^2$ product ion spectra were recorded only from ions with a charge bigger than +1 and in a data-dependent fashion in the ITMS (Ion Trap Mass Spectrometry). All relevant, individual MS settings (resolution, scan rate, scan range, AGC, ion acquisition time, charge states, isolation window, fragmentation type and details, cycle time, number of scans performed, and various other settings) for the individual experiments can be found in Supplementary File Sample_Legend_and_LC-MS_Settings, Section "MS_Settings").

**Protein identification using MaxQuant and Perseus.** RAW spectra were submitted to an Andromeda search in MaxQuant[83] (version 1.6.10.43) using the default settings. Label-free quantification and match between runs was activated. MS/MS spectra data were searched against the UniProt *P. chrysosporium* (Phanerochaete_chrysosporium_Uniprot_210114.fasta; 430 entries) and Joint Genome Institute *P. chrysosporium* (Phanerochaete_chrysosporium_JGI_210114.fasta)[42] (best-filtered model, 13602 entries) database. All searches included a contaminants database (as implemented in MaxQuant, 246 sequences). The contaminants database contains known MS contaminants and was included to estimate the level of contamination. Andromeda searches allowed the oxidation of methionine residues (16 Da), acetylation of the protein N-terminus (42 Da) as dynamic modifications, and the static modification of cysteine (57 Da, alkylation with IAM). Enzyme specificity was set to "Trypsin/P". The instrument type in Andromeda searches was set to Orbitrap and the precursor mass tolerance was set to ±20 ppm (first search) and ±4.5 ppm (main search). The MS/MS match tolerance was set to ±0.5 Da. The peptide spectrum matched FDR and the protein FDR were set to 0.01 (based on the target-decoy approach). The minimum peptide length was seven amino acids. For protein quantification, unique and razor peptides were allowed. Modified peptides with dynamic modifications were allowed for quantification. The minimum score for modified peptides was 40. Match between runs was enabled with a match time window of 0.7 min and match ion mobility window of 0.05 min[84]. Further data analysis and filtering of the MaxQuant output was done in Perseus v1.6.2.3.[85] Label-free quantification (LFQ) intensities were loaded into the matrix from the proteinGroups.txt file and potential contaminants, as well as hits from the reverse database and hits only identified by peptides with a modification site, were removed. Related biological replicates were combined into categorical groups to allow comparison of different treatments or culture media. The data were transformed to the log$_2$-scale and only those proteins that were found in two of three or three of four replicates, respectively, were investigated separately. Prior to quantification, missing values were imputed from a normal distribution (width 0.3, downshift 1.8).

In labeling experiments, the log$_2$-fold enrichment of protein groups by FP-alkyne or JJB111 was calculated based on a two-sided Student's $t$-test (permutation-based FDR: 0.05, s = 0.1, 250 randomizations) compared to the DMSO control. Proteins with a log$_2$-fold change >2 were considered significantly enriched and all proteins with a positive fold change were plotted against their numerical order. To examine the effect of the competitor pretreatment on protein enrichment, a two-sided Student's $t$-test (permutation-based FDR: 0.05, s = 0.1, 250 randomizations) was performed to calculate the difference in protein abundance between noncompetitive and pretreated probe-labeled samples and the statistical significance of the fold change. The log$_2$-fold change for samples preincubated with the corresponding competitors compared to noncompetition samples labeled with the respective probe was plotted against the –log $p$ value. Proteins with a reduction of >75% in their abundance and a $p$ value < 0.01 were considered as primary hits, while proteins with a $p$ value < 0.05 were reported as secondary hits. The protein ID was reported as either JGI ID or Uniprot ID.

**Isolation of *P. chrysosporium* mRNA, cDNA synthesis, and cloning.** For the synthesis of cDNA from *P. chrysosporium* DSM 1556, suspension cultures were grown on a minimal medium with beech wood chips for 5 days. Afterward, cells were separated from the beech wood chips by decantation and collected by centrifugation (6000×$g$, 4 °C, 30 min) and resuspended in 5 mL buffer (10 mM TRIS, pH 8). About 500 µL of cells were then transferred into a 0.1/0.5 mm bashing beads vial (Zymo Research, USA) and lysed in a bead beater (Precellys 24, VWR, USA). Afterward, cell debris was removed by centrifugation (16,000×$g$, 2 min) and RNA was isolated from 300 µL of lysed cells using the Monarch total RNA isolation kit (New England Biolabs, USA), by mixing with 300 µL of RNA lysis buffer. After RNA isolation, a cDNA library of *P. chrysosporium* DSM 1556 was synthesized using the SMARTer PCR cDNA synthesis kit (Takara Bio Europe, France). cDNA was stored at −20 °C and used for the amplification and sequencing of targeted genes. The *P. chrysosporium* gene *phchr2|126075* was amplified without introns

from cDNA using Q5® polymerase (New England Biolabs, USA) and the following gene-specific primers (Eurofins Genomics, Germany) 5′-ATGAGGTTGA-CATGTCCC-3′ and 5′-ACCTCCAATTCCTCGG-3′. The resulting PCR product was then used as a template for the amplification of *phchr2|126075* containing additional specific restriction sites for the pKLAC2 vector using the following primers: 5′-GAGGAGCATATGATGAGGTTGACATGTCCC-3′ and 5′-GAG-GAGCTCGAGACCTCCAATTCCTCGG-3′ (*Nde*I and *Xho*I restriction sites underlined). Afterward, the PCR products were purified using the Wizard® SV Gel and PCR clean-up kit (Promega, USA). *Phchr2|126075* was cloned into the pKLac2 vector (Novagen, USA), after restriction digest of the purified PCR products and the empty vector with the respective restriction enzymes (NEB, USA). The restricted PCR product and vector were used in a molar ratio of 1:4 for ligation using T4 DNA ligase (New England Biolabs, USA) at 16 °C overnight. *E. coli* DH5α cells (Novagene, USA) were transformed with the obtained constructs and the presence of successfully cloned genes was confirmed by sequencing using the gene-specific primers above. pKLAC2:*phchr2|126075* was then transformed into *K. lactis* GG799 following the manufacturer's instructions (New England Biolabs, USA). Correct integration of *phchr2|126075* was confirmed by PCR using the supplied integration primers. Transformed clones were inoculated in 2 mL of YPGlu medium to test for secretion of Phchr2|126075. Expression clones were isolated and resuspended in 250 µL of sterile 20% (v/v) glycerol and stored for further use at −80 °C. For expression of *phchr2|302168* from *P. chrysosporium* and *WP_074995790* from *Streptomyces misionensis* in *E. coli* Rosetta DE3, the coding sequences of both genes were synthesized by BioCat (Germany) without secretion signals for cloning into the pET20b-vector (with C-terminal His-tag). For heterologous overexpression, *E. coli* Rosetta DE3 was freshly transformed with the corresponding plasmid.

**Heterologous overexpression of *phchr2|126075*, *phchr2|2915237*, and *WP_074995790*.** Recombinant production of Phchr2|126075 was performed in *K. lactis* GG799 (New England Biolabs, USA). After transformation, the culture supernatant was collected by centrifugation (4000 × $g$, 30 min, 4 °C) and screened for the clone with the highest activity against *p*NP-acetate (50 mL culture volume). To this end, 100 µL of the supernatant was incubated with 100 µL of 50 mM TRIS pH 8 and 400 µM *p*NP-acetate, and the release of *p*-NP in a 96-well plate (BRANDplates®, BRAND, Germany) using a Tecan infinite M200 plate reader (Tecan Trading AG, Switzerland). For protein expression, 50 mL of culture was inoculated with 1% (v/v) of a pre-culture and incubated for 3 days at 30 °C. Afterward, cells were centrifuged (4000 × $g$, 30 min, 4 °C), and the supernatant passed through a 0.45 µm filter (Rotilabo® syringe filter, Carl Roth, Germany) before being used for determination of esterase activity of Phchr2|126075.

For the production of WP_074995790, a freshly inoculated *E. coli* Rosetta DE3 [pET20b::*WP_074995790*] culture in terrific broth (TB) medium (500 mL), supplemented with 150 µg mL$^{-1}$ ampicillin and 50 µg mL$^{-1}$ chloramphenicol, was grown to an OD$_{600}$ of 0.8 at 37 °C under constant shaking (180 rpm). Protein expression was induced by the addition of 1 mM isopropyl-β-D-thiogalactopyranoside (IPTG). The cells were then incubated at 18 °C for another 16 h, collected by centrifugation (6000 × $g$, 20 min, 4 °C), and resuspended in 10 mL of buffer A (50 mM TRIS-HCl pH 7.8, 200 mM KCl, 10 mM imidazole) per gram of wet weight. Cell lysis was performed by sonication using a UP 200 S sonicator (Hielscher Ultrasonics GmbH, Germany) for 3 × 7 min (50% amplitude, 0.5 s$^{-1}$) under constant cooling followed by centrifugation (14,000 × $g$, 60 min, 4 °C). For further purification of WP_074995790, the cleared lysate was passed through a 0.45 µm filter (Rotilabo® syringe filter, Carl Roth, Germany) and applied onto a 5 mL Ni-IDA column (Cytiva, USA) equilibrated with buffer A at a flow speed of 5 mL min$^{-1}$. After washing with buffer A (20 column volumes), elution was performed with a linear gradient of buffer B (50 mM TRIS pH 7.8, 200 mM KCl, 400 mM imidazole). The elution buffer was exchanged for storage buffer (50 mM TRIS pH 8.0, 20 mM KCl, 10% (v/v) glycerol) using Amicon® centrifugal filter devices (50 kDa cutoff, Merck, Germany) by repeated concentration and dilution. For storage, proteins were flash-frozen in liquid nitrogen and stored at −80 °C.

A synthetic gene encoding Phchr2|2915237 was ordered as a gBlock from IDTdna (USA), integrated into the genome of *Aspergillus oryzae* and expressed as an extracellular enzyme as described elsewhere[86]. A C-terminal his-tag (6xHis) was added to ease downstream purification. The fermentation broth was sterile filtrated and 500 mM NaCl was added and adjusted to pH 7.5 by the addition of NaOH. The sample was loaded onto a Ni-Sepharose™ 6 Fast Flow column (GE Healthcare, USA) equilibrated in 50 mM HEPES, pH 7.5 with 500 mM NaCl (buffer A). After loading, the column was washed with 10 column volumes of buffer A, and bound proteins were eluted with 500 mM imidazole in buffer A. The fractions containing the enzyme were pooled and applied to a Sephadex™ G-25 (medium) column equilibrated and eluted in 100 mM HEPES pH 7.5. Fractions were analyzed by SDS-PAGE, and fractions containing the enzyme were combined.

**Activity assay for recombinant CE1.** Esterase activity was confirmed by measuring the release of *p*NP from *p*NP-acetate at 410 nm in either a discontinuous or continuous assay. To confirm the pH optimum of Phchr2|126075 1.08 µg of protein was incubated in 50 mM phosphate citrate buffer at a pH range from 5.5 to 8 with 25 mM of *p*NP-acetate in a total volume of 500 µL for 10 min at 35 °C.

Afterward, the reactions were stopped by the addition of 500 μL of 0.5 M sodium carbonate and the absorption of samples was determined at 410 nm. The production of $p$NP was then calculated using the established extinction coefficient of $16.1\,\text{mM}^{-1}\,\text{cm}^{-1}$ for $p$NP[87]. For determination of the temperature optimum, Phchr2|126075 was incubated in 50 mM TRIS pH 8, 20 mM KCl and release of $p$NP from $p$NP-acetate was continuously determined in a temperature range from 30 to 70 °C in a total volume of 500 μL. For the determination of kinetic constants, Phchr2|126075 was incubated with different concentrations of $p$NP-acetate at 40 °C in an aqueous solution containing 50 mM TRIS pH 8 and 20 mM KCl. Initial velocities of each reaction were taken and used for the calculation of activities. All enzyme assays were performed in triplicate.

**Activity assay for recombinant glycoside hydrolases**. The activity of Phchr2|2915237 and WP_074995790 was established against different polysaccharides and artificial $p$NP-conjugates. The pH and temperature optimum of Phchr2|2915237 were determined by incubating 0.5 μg of an enzyme with 200 μM of $p$NP-Glc or $p$NP-Gal for 10 min in 500 μL phosphate citrate buffer[88] at a pH range of 3–8 at 30 °C or a temperature range of 30–70 °C at pH 5, respectively. After incubation, reactions were stopped by the addition of 500 μL of 0.5 M sodium carbonate and cleavage of $p$NP-Glc was followed by determining the release of $p$NP at 410 nm as described above. For measuring the substrate specificity of Phchr2|2915237 and WP_074995790 against $p$NP-conjugates, either 200 μM of $p$NP-Ara, $p$NP-GlcNAc, $p$NP-Gal, and $p$NP-Man were incubated with 10 μg of Phchr2|2915237 or 11.2 μg of WP_074995790 or 200 μM of $p$NP-Xyl and $p$NP-Glc were incubated with 0.4 μg of Phchr2|2915237 or 11.2 μg of WP_074995790 as described above. Kinetic measurements against nitrophenyl substrates were performed in a discontinuous assay by incubating different concentrations of $p$NP-Glc and $p$NP-Xyl with 0.5 μg of Phchr2|2915237 while stopping the reaction after 1, 2, 5, and 10 min, respectively. Afterward, the initial velocities of the reaction were used to calculate the specific activity of Phchr2|2915237 towards nitrophenol substrates.

For measuring the substrate specificity of Phchr2|2915237 against xyloglucan, galactomannan, curdlan, CMC, xylan, and lichenan, 1 μg of enzyme was incubated with 0.5 % (w/v) of the respective polysaccharides in 500 μL of in citric acid phosphate buffer pH 5 for 1, 2, 5, and 10 min. Afterward, 500 μL of the DNSA solution (10 g L$^{-1}$ DNSA, 2 mL L$^{-1}$ of 0.05 g L$^{-1}$ sodium sulfite, 200 g L$^{-1}$ potassium sodium tartrate, and 10 g L$^{-1}$ NaOH) was added and subsequently incubated at 100 °C for 15 min. Samples were then cooled on ice for 15 min and centrifuged (10,000×$g$, 4 °C for 30 min) before 250 μL of the supernatant was transferred into a 96-well plate (BRANDplates®, BRAND, Germany). The release of reducing sugars was followed at 575 nm using a Tecan infinite M200 plate reader (Tecan Trading AG, Switzerland) and determined using a calibration curve based on D-glucose. Background absorption due to abiotic substrate degradation and the addition of protein solutions was determined and subtracted from the absorption of the samples. Glutaminase activity was tested using L-gamma-glutamyl-$p$NP as described before[89] or by monitoring the formation of glutamate from glutamine by glutamate dehydrogenase (Merck, Germany) via the reduction of NAD$^+$. To check for the formation of glucose or xylose during polysaccharide degradation, samples were incubated as described above for 4 h. Afterward, samples were centrifuged at 10,000×$g$, 4 °C for 30 min, and 200 μl of supernatant was incubated with either 2 U of glucose dehydrogenase (Sigma Aldrich, USA) or xylose dehydrogenase (Megazyme, Ireland) and 5 mM of NAD$^+$ in 50 mM TRIS pH 8 and the formation of glucose and xylose was then measured by determining the reduction of NAD$^+$ to NADH. All enzyme assays were performed in triplicate.

*Analysis of hydrolysis products by thin-layer chromatography (TLC)*. Enzymatic reactions with 0.5 % (w/v) lichenan or beech wood xylan were performed under the same conditions as used for the DNSA assay with 10 μg of Phchr2|2915237. Lichenan and xylan were incubated with Phchr2|2915237 as described above, and samples were removed after 1, 2, and 4 h. Afterward, 2.5 μl of the hydrolysis products and control solutions were applied to aluminum sheet silica gel 60/ kieselguhr F254 plates ((20 cm × 20 cm, Merck, Germany) and separated at room temperature with ethyl acetate, methanol and H$_2$O (68:23:9, v/v/v) as solvent. Plates were dried and products were visualized by treating the plates with a KMnO$_4$ staining solution (1.5 g KMnO$_4$, 10 g K$_2$CO$_3$, and 1.25 mL 10% aq. NaOH in 200 mL H$_2$O) and incubation at room temperature for 30 min.

**Reporting summary**. Further information on research design is available in the Nature Research Reporting Summary linked to this article.

## Data availability

The mass spectrometry proteomics data for the digestions have been deposited to the ProteomeXchange Consortium via the PRIDE[90] partner repository (https://www.ebi.ac.uk/pride/archive/) with the dataset identifier PXD030618. All data generated during this study are included in this published article (and its supplementary information files). All source data underlying the graphs and charts can be found in Supplementary Data 5. Raw datasets generated during the current study are available upon request.

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

## Acknowledgements
Funding by the Mercur Mercator Research Center Ruhr (Pr-2017-0020, to D.B., B.S., and M.K.) as well as the DFG (INST 20876/322-1, to M.K. and F.K.) is greatly acknowledged.

## Author contributions
C.B., D.B., B.S., and M.K. conceived and designed the study; C.S., L.S., G.H., K.J., and F.W. performed and analyzed all chemical biology experiments. C.S. and K.J. conducted growth studies and protein expression. C.S. performed enzyme characterizations. H.S.O. provided the activity-based probes. L.S., C.S., G.H., and F.K. performed mass spectrometry and related data analysis. C.S., L.S., B.S., and M.K. wrote the manuscript.

## Funding

## Competing interests
The authors declare no competing interests.
