## [Peer Review File · Communications Biology]

Reviewers' comments:

Reviewer #1 (Remarks to the Author):

The authors devise a novel strategy for fishing out biomass-active enzymes by activity as opposed to homology searching. The identification of novel enzymes that do not share sequence similarity with the known pool of enzymes is an important goal. Identifying novel enzymes by sequence similarity alone is helpful but only goes so far and the strategy of the technique itself almost precludes identification of truly novel sequences. Thus, the goal of an unbiased means to identifying enzyme activity in the absence of sequence identity is a worthy pursuit.

There are obviously a ton of glycosyl hydrolase families. It is mentioned in line 99 that KY371 and JJB111 are "GH-targeting ABPs." Is there any known specificity here? Does it target them all? That seems unlikely to have been tested. Has any GH-specificity been determined or has it just been shown to react with some GHs? This is important as it shows the maximum opportunity space for this ABP to find enzymes. While I know less about the serine hydrolases, I guess my question would be the same there.

Related to the above question, in the case of JJB111, twelve GHs were identified in the soluble and 7 were identified in the insoluble. It would be helpful to know explicitly what percent of the total GHs were recovered. It was hard to tell if this is able to be ascertained from your supplemental figure. Are there any hypothesized trends as to why these specific 12 were identified? I'm assuming there were more active GHs in the media that didn't react with JJB111.

I think an excellent experiment would be to test your GH/SH-reactive ABPs with known cellulases and serine hydrolases, either commercially purchased or generated to understand the reaction more clearly.

If it is a bit random as to what is reacting with the different ABPs, then it causes me concern about the relevance of the assay. What exactly is being identified except for some random GHs, and SHs that interact with your specific ABPs? What is the relevance, and why is this a good means to find enzymes? To me this remains the largest question mark I have about the method and its impact as a screening method moving forward.

A major touted advantage of this approach is that it would enable the identification of enzymes that weren't sequence-predicted to be GHs. While there was an attempt to categorize one of these (Phchr2|3002168) its expression failed so it couldn't be tested. A sequence-based homolog of this enzyme was tested and showed modest activity. While this may point to the fact that the Phanaerochaete enzyme may have been a GH, it is certainly not conclusive. And is instead preliminary.

On the Substrate-bound enzyme fraction, it would be good to know how effective your extraction was. While I realize you need to extract active enzymes for your assay to work, it would be good to see data comparing your detergent-extracted fraction with a more harshly-extracted fraction (by SDS-PAGE or proteomics, for example).

It was good to see that when you purified the enzymes that they possessed some activity on model substrates, but it is difficult to ascertain the impact here. Does anything make these enzymes stand out beyond the fact that they're active on model substrates?

Minor notes:

There is a decent amount of jargon and abbreviations peppered throughout that should be defined at first mention. For example: "warhead," "CE15,"

The grammar and syntax could use some improvement, but overall message is clear. There are a few words that seem ubiquitously misused. I *think* that "homologues" is being used instead of "homologous" in some cases: line 159- "...as homologues proteins..." line 269 "...secreted Phchr2|3002168-homologues proteins..." etc.

There are a lot of very specific experimental details in the results that are not helpful in understanding the experiment or the results and would be better moved to M&M to streamline the communication of experimental results.

Line 106: This seems an overly generalized statement and is not referenced. It sounds as if serine hydrolases are essential to hemicellulose degradation, but I am unaware of this. Please provide additional rationale as to the inclusion of serine hydrolases as a target. Are you specifically thinking about acetyl xylan esterases?

Line 123: "inorganic" instead of "anorganic"?

Line 141: "Sentence is confusingly written. Might suggest switching to "Phchr2-126075 displays high sequence similarity to a previously studied acetyl xylan esterase of *P. chrysosporium*, and its labeling was competed by pre-treatment with the esterase inhibitor paraoxon"

Line143: Note that "CE" refers to a carbohydrate esterase.

Line 144: Please explain this connection in more detail. Are all CE15s 4-O-methyl-glucuronoyl methyl esterases? How are you coming to this conclusion? CAZY?

Line 180-183: How did you separate fungal biomass from beechwood biomass?

Line 181: How do you know how efficient this method was in removal of enzymes? How do you know they were active?

Reviewer #2 (Remarks to the Author):

The authors have described the use of activity-based protein profiling to characterize fungal serine hydrolases and CAZymes. Several prior studies have evaluated lignocellulose deconstruction by probe profiling, but this report does make an advance in profiling substrate-bound fungi. To identify new enzymes as potential biocatalysts it is important to evaluate those that are expressed while the fungi is actively growing on substrate.

Overall, the work is of interest, but suffers from lack of completeness.

Major points:

1. Substrate bound fungi deconstructs complex lignocellulose polymer into smaller substrates. These substrates can also serve to compete against probe labeling. Given the genome, what enzymes are likely being missed as a function of substrate competition? See number 2 as follow up.
2. Table 2 targets - I'm surprised by the lack of glucosidases, hemicellulases, and cellulases. I would anticipate all of these to be expressed while the organism is growing on substrate. Explanation should be provided.
3. An explanation on the limitation of the ABPP approach for biocatalyst screening should be given.

4. The study would be much enhanced to better simulate some of the pretreatment approaches currently being used. For instance, could the beech substrate be pretreated enzymatically or thermochemically and then your analysis performed on fungi growing on the treated material? It would be very relevant to identify potential enzymes as biocatalysts within conditions relevant to industry.

5. How does growth stage on substrate impact expression of functional enzymes? This question points to the somewhat limited approach taken in this manuscript to identify lignocellulose degrading-relevant enzymes as biocatalysts. Fungal growth stage is critical to enzyme expression.

Minor points:

- The interchange of "biocatalyst" and "enzyme" is a bit confusing. It would be helpful right from the onset to define ABPP-identified relevant enzymes as the biocatalysts
- Line 57, change to "For example, ABPP can be used...."
- Line 72, remove "however so far mostly"
- Line 75, remove "therefore"
- Line 170, remove "also"
- Line 173 and 179, remove "however"
- Line 217, remove "therefore"
- Line 279, add "we" after the comma
- Line 300, remove "such"

Reviewer #3 (Remarks to the Author):

The manuscript submitted by Schmerling and colleagues reports the utilization of the well-established "activity-based protein profiling" methodology to evaluate the enzymatic secretion potential of *Phanerochaete chrysosporium* focusing on carbohydrate-active enzymes in the context of wood decay. Using their methodology, the authors managed to identify some CAZymes and more interestingly, some DUFs that could become new CAZy families in the future.

The manuscript is well written. The study is technically sound and the methods are well described. I have only minor comments to improve the manuscript before publication.

The title should be more accurate clearly stating that their was a focus on *Phanerochaete chrysosporium*. The study is different from recent work (McGregor et al 2022 Biotech for biofuels) where a functional screening was carried out on different white rot fungi (this recent publication should be quoted). Therefore the title could be modified to : "Identification of fungal lignocellulose-degrading biocatalysts secreted by *Phanerochaete chrysosporium* via activity-based protein profiling". In the abstract, the authors should state more clearly which CAZy families were identified.

Line 85: Do you means LPMOs? LPMOs belong to the AA class.

Line 236: Regarding Phchr2|2915237, what is the CAZy GH5 subfamily? The activity observed on beta1,3/1,4 glucan (lichenan) is interesting . It could suggest action of this enzyme towards fungal cell wall rather than plant cell wall. This should be discussed in the context of wood decay. It would have been interesting to assess the activity of this enzyme on other types of glucans and identify the soluble products of the reaction. It would be nice to know whether the action of this enzyme is exo- or endo-type? Are there any reports in the literature of dual glucanase/xylanase activity?

Line 274: *S. misionensis*? *S.* stands for which genus?

Reviewer #1 (Remarks to the Author):

The authors devise a novel strategy for fishing out biomass-active enzymes by activity as opposed to homology searching. The identification of novel enzymes that do not share sequence similarity with the known pool of enzymes is an important goal. Identifying novel enzymes by sequence similarity alone is helpful but only goes so far and the strategy of the technique itself almost precludes identification of truly novel sequences. Thus, the goal of an unbiased means to identifying enzyme activity in the absence of sequence identity is a worthy pursuit.

We thank the reviewer for his/her overall positive feedback on our study.

There are obviously a ton of glycosyl hydrolase families. It is mentioned in line 99 that KY371 and JJB111 are “GH-targeting ABPs.” Is there any known specificity here? Does it target them all? That seems unlikely to have been tested. Has any GH-specificity been determined or has it just been shown to react with some GHs? This is important as it shows the maximum opportunity space for this ABP to find enzymes. While I know less about the serine hydrolases, I guess my question would be the same there.

Both probes KY371 and JJB111 have previously been used in different ABPP campaigns. They both possess an N-alkynyl-cyclophellitol aziridine moiety as a warhead and have originally been developed as broad spectrum ABPs for retaining β -glucosidases (e.g. Kallemeijn et al., *Angew. Chem. Int. Ed. Engl.* 2012, 51, 12529, ref. [49] in our manuscript). Their proteome-wide target repertoire has however also been investigated by ABPP, for example in plants in (Chandrasekar et al., *Mol. Cell. Proteom.* 2014, 13, 2787, ref. [52] in our manuscript). In this study, JJB111 labeled seven different families of retaining glycosidases (e.g. myrosinases, glucosidases, xylosidases and galactosidases). JJB111 profiling was thereby performed in *Arabidopsis thaliana* as well as *Nicotiana benthamiana* and resulted in the identification of nearly 40 different retaining glycosidases. Further experiments on JJB111-labeled peptides revealed modification of the nucleophilic glutamate, indicating labeling of the enzyme active site. In a broad-range profiling experiment, they showed that the ABP, however, had additional specificities toward xylosidases, galactosidases, glucuronidases, and glucanases. In addition, in their screening experiment, an α -L-arabinofuranosidase and heparanase were identified. Altogether, these previous studies show that a wide range of in particular retaining glycosidases can be identified with these probes.

The FP probe is a well-known broad range probe for serine hydrolases (e.g. Simon et al., *J. Biol. Chem.* 2010, 285, 11051, ref. [47] in our manuscript). Indeed, multiple previous studies have shown that the FP probe reacts with both serine proteases and metabolic serine hydrolases. Metabolic serine hydrolases are a diverse set of enzymes, including peptidases, lipases, esterases, thioesterases, and amidases. In an exemplary proteome-wide study in mouse cells and tissues by Bachovchin et al. (Bachovchin et al., *Proc. Natl. Acad. Sci.* 2010, 107, 20941, new ref. [51] in our manuscript), more than 100 FP-labeled serine hydrolases were identified. Compared with the predicted number of metabolic serine hydrolases in mice, this corresponds to roughly 82% of all serine hydrolases, emphasizing the potential of the FP probe as a broad-range probe that targets almost all serine hydrolases. However, these results indicate that there is no specificity within the enzyme class, which is indeed a limitation for our analysis.

A corresponding further note on their established target repertoire has been added to the manuscript.

Related to the above question, in the case of JJB111, twelve GHs were identified in the soluble and 7 were identified in the insoluble. It would be helpful to know explicitly what percent of the total GHs were recovered. It was hard to tell if this is able to be ascertained from your supplemental figure. Are there any hypothesized trends as to why these specific 12 were identified? I'm assuming there were more active GHs in the media that didn't react with JJB111.

In the culture supernatant as well as in the substrate-bound fraction, more than 12 GHs were identified but only 12 GHs were significantly enriched with a FC>2 compared to the DMSO control. We agree with the reviewer that our ABPP approach will not detect all present GHs. Some will not react with the ABP due to their specificity, others will not be active. In addition, it is likely that some active GHs have even reacted with the ABP but are only low abundant and thus out of the detection limit of the MS analysis.

We are however not able to provide a ratio as we cannot quantify the number of GHs that are present in the supernatant (or substrate-bound fraction). We tried to quantify the number of available GHs by a standard full proteome analysis. However, several ABPP-detected GHs were not found in the full proteome analysis. We therefore cannot make a valid assumption on how many GHs are present and on how many of them are targeted by the ABPs.

I think an excellent experiment would be to test your GH/SH-reactive ABPs with known cellulases and serine hydrolases, either commercially purchased or generated to understand the reaction more clearly.

The aim of our study has not been to identify the exact selectivity or to elucidate the reaction mechanism of the used ABPs. Indeed, for the GH probes, these issues have already been addressed in previous studies (e.g. the above-mentioned studies of Chandrasekar et al., *Mol. Cell. Proteom.* 2014, 13, 2787 or Kallemeijn et al., *Angew. Chem. Int. Ed. Engl.* 2012, 51, 12529, as well as many more). Also the FP-based SH probes have been widely studied in this regard (e.g. the above-mentioned Bachovchin et al., *Proc. Natl. Acad. Sci.* 2010, 107, 20941, but also for example Kidd et al., *Biochemistry* 2001, 40, 4005; Patricelli et al., *Proteomics* 2001, 1, 1067; Jessani et al., *Proc. Natl. Acad. Sci.* 2002, 99, 10335, and many more). We therefore do not feel that this type of experiment is still required.

We rather intended to use the already existing knowledge about the respective ABPs in order to screen complex heterogeneous microbial cultures for enzymes of biotechnological interest. By labeling with the ABPs, it is possible to pre-select for promising, in the process-active enzymes which cannot be easily achieved by other bioinformatics or 'omics'-based approaches.

If it is a bit random as to what is reacting with the different ABPs, then it causes me concern about the relevance of the assay. What exactly is being identified except for some random GHs, and SHs that interact with your specific ABPs? What is the relevance, and why is this a good means to find enzymes?

To me this remains the largest question mark I have about the method and its impact as a screening method moving forward.

As already stated above in our previous responses and as shown by many different groups from the ABPP field, the ABPs have a distinct target selectivity and thus do not react randomly. However, as in all screening approaches, we indeed do also detect “false positives” but as can be seen from Fig. 2B-C and 3B-C, the majority of identified enzymes belong to the expected target families. We therefore believe that our suggested ABPP-based approach enables a rapid and straightforward preselection of enzyme candidates for further analysis. Our approach thus allows to focus the biochemical expression and validation of enzymes to the most promising candidates.

A major touted advantage of this approach is that it would enable the identification of enzymes that weren't sequence-predicted to be GHs. While there was an attempt to categorize one of these (Phchr2|3002168) its expression failed so it couldn't be tested. A sequence-based homolog of this enzyme was tested and showed modest activity. While this may point to the fact that the Phanaerochaete enzyme may have been a GH, it is certainly not conclusive. And is instead preliminary.

Regarding Phchr2|3002168 we completely agree that our data is not completely conclusive, as the failed expression of the protein prevented a direct confirmation of its enzyme activity via a biochemical assay. We can therefore only conclude that it might be a GH. We have therefore critically revised our wording in the manuscript, emphasizing that we have only indications that Phchr2|3002168 is a GH.

However, we have strong indications that this is indeed the case. In Fig. 2C, almost all significant hits were annotated GHs, with the exception of Phchr2|3002168 being the main non-GH 'outlier'. As previous studies with JJB111 clearly describe its β -retaining GH specificity and as the bacterial sequence homologue displays GH activity, it is therefore reasonable to assume that Phchr2|3002168 is indeed also a GH. This finding is in line with the observation by Arntzen et al. (Arntzen et al., *Sci. Rep.* 2020, 10, 20267, ref. [69] in our manuscript) that reports that these kind of multidomain DUF proteins are secreted by different fungi when grown on lignocellulose.

However, and in accordance with the comment of the reviewer, we again want to emphasize that we are aware that we have not directly confirmed the GH enzyme activity of Phchr2|3002168. A conclusive confirmation will require further studies into this direction.

On the Substrate-bound enzyme fraction, it would be good to know how effective your extraction was. While I realize you need to extract active enzymes for your assay to work, it would be good to see data comparing your detergent-extracted fraction with a more harshly-extracted fraction (by SDS-PAGE or proteomics, for example).

We thank the reviewer for this suggestion. We are aware that a detachment using a non-denaturing detergent such as DDM is a mild method and may result in missing some potential target enzymes. To evaluate if a 'stronger' detergent might detach more proteins, we performed a proteome experiment in which we compared the number of protein identifications after substrate detachment with either DDM or SDS.

The analysis of the substrate-bound fraction resulted in the identification of 2608 proteins, of which 1994 proteins were found either with DDM or SDS, whereas 142 proteins were exclusively found in the DDM approach and 472 in the SDS approach. In contrast, a proteome analysis of secreted proteins revealed that 245 proteins were identified across both approaches. Interestingly, however, 34 proteins were found exclusively after the DDM detachment and 22 proteins exclusively using the workflow with SDS. Altogether, this suggests that DDM detachment of the SBF, as we have performed, should be the preferable method as the majority of proteins are identified after detachment with both methods. The SDS procedure however may trigger cell lysis, resulting in a 'contamination' of the fraction with intracellular proteins. Further on, the SDS procedure will inactivate enzymes and thus prevent the identification of active enzymes by ABPP, which is why we have not included the results of our SDS solubilization experiment in the manuscript.

It was good to see that when you purified the enzymes that they possessed some activity on model substrates, but it is difficult to ascertain the impact here. Does anything make these enzymes stand out beyond the fact that they're active on model substrates?

We think that the reviewer refers here to Fig. 4B? Biochemical assays were performed to investigate the activity and specificity of the heterologously expressed enzymes. These assays were based on the usage of model substrates.

However, for Phchr2|2915237, activity towards complex substrates such as xylan and lichenan was also shown. Xylan is a type of hemicellulose and lichenan is a glucan. For the enzyme WP_074995790, however, we could indeed not demonstrate any activity towards biologically more relevant substrates but only against selected model substrates.

Minor notes:

There is a decent amount of jargon and abbreviations peppered throughout that should be defined at first mention. For example: "warhead," "CE15,"

Abbreviations are now conclusively defined at their first mention. We also tried to define all types of jargon such as "warhead" at their first appearance in the text.

The grammar and syntax could use some improvement, but overall message is clear. There are a few words that seem ubiquitously misused. I *think* that "homologues" is being used instead of "homologous" in some cases: line 159- "...as homologues proteins..." line 269 "...secreted Phchr2|3002168-homologues proteins..." etc.

Changed accordingly.

There are a lot of very specific experimental details in the results that are not helpful in understanding the experiment or the results and would be better moved to M&M to streamline the communication of experimental results.

We carefully went through the manuscript, trying to streamline the experimental results as suggested by the reviewer. This has resulted in the removal of some 'experimental details' in the results section (the corresponding experimental descriptions have already been present in the M&M section). We hope that the reviewer finds the results section now easier to read.

Line 106: This seems an overly generalized statement and is not referenced. It sounds as if serine hydrolases are essential to hemicellulose degradation, but I am unaware of this. Please provide additional rationale as to the inclusion of serine hydrolases as a target. Are you specifically thinking about acetyl xylan esterases?

Yes, we are referring here to acetyl xylan esterases that play an important role in hemicellulose degradation. We have added a corresponding review to the statement (Sista Kameshwar & Qin, Mycology 2018, 9, 273, new ref. [54] in our manuscript) and have rewritten the sentence to make better clear that we talk here about acetyl xylan esterases (and not all types of serine hydrolases).

Line 123: "inorganic" instead of "anorganic"?

Changed accordingly.

Line 141: "Sentence is confusingly written. Might suggest switching to "Phchr2-126075 displays high sequence similarity to a previously studied acetyl xylan esterase of *P. chrysosporium*, and its labeling was competed by pre-treatment with the esterase inhibitor paraoxon"

Changed accordingly.

Line143: Note that "CE" refers to a carbohydrate esterase.

We are sorry for this error and changed it accordingly.

Line 144: Please explain this connection in more detail. Are all CE15s 4-O-methyl-glucuronoyl methyl esterases? How are you coming to this conclusion? CAZY?

According to the classification of CAZY, all CE15s are 4-O-methyl-glucuronoyl methyl esterases. The classification was created following the paper by Li et al. (Li et al., FEBS Lett. 2007, 581, 4029).

Line 180-183: How did you separate fungal biomass from beechwood biomass?

After substrate detachment of secreted adherent enzymes, the supernatant was transferred to a new reaction vessel using a pipette. We recognize that we cannot make an exact separation of the fungal and wood biomass here. However, in LC-MS/MS analysis, we use only the fungal database, which

allows us to identify adherent fungal proteins. We also focused only on enzymes with secretion signals. The corresponding experimental procedure is described in the Materials and Methods section of the Supplementary Information.

Line 181: How do you know how efficient this method was in removal of enzymes? How do you know they were active?

Again, we cannot really quantify the efficiency of our removal. However, we are able to detect active enzymes, including some that we did not find with the “standard” ABPP approach in the supernatant; this shows that the ABPP approach is feasible and works at least partially. As ABPP is known to only label active enzymes (this has been shown in numerous ABPP publications), we anticipate that all detected enzymes are indeed active. In addition, we had also performed activity measurements of the SBF which showed activity towards model substrates and complex polysaccharides, thereby confirming that this fraction must have contained active enzymes.

Reviewer #2 (Remarks to the Author):

The authors have described the use of activity-based protein profiling to characterize fungal serine hydrolases and CAZymes. Several prior studies have evaluated lignocellulose deconstruction by probe profiling, but this report does make an advance in profiling substrate-bound fungi. To identify new enzymes as potential biocatalysts it is important to evaluate those that are expressed while the fungi is actively growing on substrate.

Overall, the work is of interest, but suffers from lack of completeness.

We thank the reviewer for his/her overall positive feedback on our study.

Major points:

1. Substrate bound fungi deconstructs complex lignocellulose polymer into smaller substrates. These substrates can also serve to compete against probe labeling. Given the genome, what enzymes are likely being missed as a function of substrate competition? See number 2 as follow up.

We thank the reviewer for this suggestion. It is quite likely that there may be a competitive effect due to smaller degradation products of the substrate in course of the labeling reaction. However, the enzymes missed in the analysis due to substrate competition cannot be identified from the genome alone. In addition, it should be mentioned that ABPs form a covalent bond with the target enzyme, whereas the substrate or smaller degradation products bind non-covalently and transiently. Therefore, by incubating for 1 h with the probe, it can be assumed that only a very small fraction of active enzymes cannot be enriched due to this effect.

2. Table 2 targets - I'm surprised by the lack of glucosidases, hemicellulases, and cellulases. I would anticipate all of these to be expressed while the organism is growing on substrate. Explanation should be provided.

All used ABPs have a distinct target selectivity. The used GH ABP probes JJB111 and KY371 display specificity towards retaining β -glucosidases and therefore mainly enzymes of this class were enriched and listed in the tables (see for example Chandrasekar et al., *Mol. Cell. Proteom.* 2014, 13, 2787, ref. [52] of our manuscript, for the proteome-wide target selectivity of JJB111 in plants; please see also our answers to reviewer #1). In particular hemicellulases and cellulases are however, despite their undoubted biotechnological relevance, not well-targeted by the used probes. These enzymes require different ABPs that indeed have been developed (e.g. Schöder et al., *ACS Cent. Sci.* 2019, 5, 1067, ref. [24] in our manuscript; Chen et al., *J. Am. Chem. Soc.* 2021, 143, 2423, ref. [32] in our manuscript; Chauvigné-Hines et al., *J. Am. Chem. Soc.* 2012, 134, 20521, ref. [30] in our manuscript).

We however feel from this comment that we might not have well-explained the target selectivity of ABPs in the ABPP technology. We have therefore added one further sentence to the introduction section to make clear that the used ABP “defines” which target enzymes can be identified.

3. An explanation on the limitation of the ABPP approach for biocatalyst screening should be given.

We thank the reviewer for this comment. We are aware that our ABPP approach has some limitations and have tried to present them (mainly in the discussion section) of our study. We have identified the following limitations of the ABPP approach:

We agree that we can only study a certain set of proteins due to our fixed cultivation conditions. For example, it should be noted here that we only examined the cultures after a fixed growth time of 5 days. By shortening or lengthening the cultivation time, the composition of secreted proteins will change and thus by only changing this parameter alone, we should be able to identify different target proteins. In this study, however, we focused on the establishment of the workflow to enable studies on the detection of biocatalysts in heterogeneous microbial cultures and, in particular, of substrate-bound enzymes which are typically not addressed in classical analyses. We believe that these workflows can be used also by other groups for enzyme identification with other microbial cultures.

Furthermore, it must also be emphasized that we only used two different ABP types. An expansion of the probe library with ABPs targeting different enzyme classes would enlarge the target enzyme portfolio.

As noted by reviewer #2 in his first remark, substrate competition may represent an issue during ABPP labeling.

Finally, it has also to be kept in mind that all ABPs (in different extents) also display non-specific labeling, i.e. label non-target enzymes. Accordingly, the ABPP approach only represents a pre-selection procedure and requires subsequent bioinformatic analyses and the verification of target hits via expression and purification.

4. The study would be much enhanced to better simulate some of the pretreatment approaches currently being used. For instance, could the beech substrate be pretreated enzymatically or thermochemically and then your analysis performed on fungi growing on the treated material? It would be very relevant to identify potential enzymes as biocatalysts within conditions relevant to industry.

See our response to point 5 (below).

5. How does growth stage on substrate impact expression of functional enzymes? This question points to the somewhat limited approach taken in this manuscript to identify lignocellulose degrading-relevant enzymes as biocatalysts. Fungal growth stage is critical to enzyme expression.

Both remarks 4 and 5 are great suggestions and we would like to address these points in future studies. Growth time plays a large role in the expression and secretion of lignocellulose degrading proteins and in some preliminary experiments we already observed large changes in the labeling of secreted proteins dependent on fungal growth time. As suggested, we imagine that the same holds true for the pretreatment of growth substrate via different methods. In analogy, other sources of lignocellulose biomass such as corn stalks could also be used. Indeed, there are numerous possibilities to change growth conditions and probes used, and all these factors will influence which enzymes can be identified with our generic ABPP approach.

In the present study, we however focused on the establishment and validation of this generic ABPP approach which we believe is an important finding by itself. We aim to report its application to diverse settings in future, separate studies.

Minor points:

- The interchange of "biocatalyst" and "enzyme" is a bit confusing. It would be helpful right from the onset to define ABPP-identified relevant enzymes as the biocatalysts

We have added a corresponding statement to the main text.

- Line 57, change to "For example, ABPP can be used...."

Changed accordingly.

- Line 72, remove "however so far mostly"

Removed accordingly.

- Line 75, remove "therefore"

Removed accordingly.

- Line 170, remove "also"

Removed accordingly.

- Line 173 and 179, remove "however"

Removed accordingly.

- Line 217, remove "therefore"

Removed accordingly.

- Line 279, add "we" after the comma

Added accordingly.

- Line 300, remove "such"

Removed accordingly.

Reviewer #3 (Remarks to the Author):

The manuscript submitted by Schmerling and colleagues reports the utilization of the well-established "activity-based protein profiling" methodology to evaluate the enzymatic secretion potential of *Phanerochaete chrysosporium* focusing on carbohydrate-active enzymes in the context of wood decay. Using their methodology, the authors managed to identify some CAZymes and more interestingly, some DUFs that could become new CAZy families in the future.

The manuscript is well written. The study is technically sound and the methods are well described. I have only minor comments to improve the manuscript before publication.

We thank the reviewer for his/her overall positive feedback on our study.

The title should be more accurate clearly stating that there was a focus on *Phanerochaete chrysosporium*. The study is different from recent work (McGregor et al 2022 Biotech for biofuels) where a functional screening was carried out on different white rot fungi (this recent publication should be quoted).

Therefore the title could be modified to : "Identification of fungal lignocellulose-degrading biocatalysts secreted by *Phanerochaete chrysosporium* via activity-based protein profiling".

We apologize for not citing this important study which has now been added to the reference list (now cited as new ref. [35] in our study). We also changed the title of our manuscript as suggested.

In the abstract, the authors should state more clearly which CAZy families were identified.

We have extended the abstract, now including a better overview on the identified enzyme classes.

Line 85: Do you mean LPMOs? LPMOs belong to the AA class.

Yes, we wanted to refer to LPMOs. We have rewritten the sentence accordingly.

Line 236: Regarding Phchr2|2915237, what is the CAZy GH5 subfamily? The activity observed on beta1,3/1,4 glucan (lichenan) is interesting. It could suggest action of this enzyme towards fungal cell wall rather than plant cell wall. This should be discussed in the context of wood decay. It would have been interesting to assess the activity of this enzyme on other types of glucans and identify the soluble products of the reaction. It would be nice to know whether the action of this enzyme is exo- or endo-type? Are there any reports in the literature of dual glucanase/xylanase activity?

We thank the reviewer for these interesting suggestions. Phchr2|2915237 is predicted to belong to the GH5 subfamily 9. Originally, we did not consider Phchr2|2915237 to be involved in cell wall metabolism, however, as the reviewer suggests, this could certainly be the case.

To check for exo-glucanase activity, we therefore made an additional experiment and analyzed the hydrolysis products generated by Phchr2|2915237 via thin-layer chromatography (new Supplemental Figure 4). However, in our enzyme assays (hydrolysis of lichenan and xylan), we were not able to observe the release of the monosaccharide products xylose or glucose; instead, we only observed the formation of polysaccharides of unknown length in a time dependent manner. In addition, we confirmed that indeed no glucose/xylose monosaccharides were released during our reaction by an additional glucose as well as xylose dehydrogenase coupled assay. Based on these observations, we conclude that Phchr2|2915237 is most probably an endo-cleaving glucanase.

Regarding the potential involvement in plant cell wall degradation: Similar enzymes from *Aspergillus* species are suggested to be involved in the utilization of extracellular glucans. In light of our results, we therefore believe that Phchr2|2915237 is also involved in the degradation of extracellular glucans, since its activity would allow for utilization of xylan and the protein was secreted under growth on lignocellulose. However, we cannot conclusively determine the exact physiological function of Phchr2|2915237. Dual glucanase/xylanase activity was not reported in this GH5 subfamily before, however only a few proteins of this subfamily have been intensively characterized. For other GH families, a dual xylanase/glucanase activity is however not uncommon (see for example Girfoglio et al., J. Bacteriol. 2012, 194, 5091 or Gloster et al., J. Biol. Chem. 2007, 282, 19177).

We have added a new section to the performed experiments as well as potential role of Phchr2|2915237 to the manuscript text.

Line 274: *S. misionensis*? *S.* stands for which genus?

S. misionensis stands for *Streptomyces misionensis*. We have now written the complete genus on its first mention.

REVIEWERS' COMMENTS:

Reviewer #2 (Remarks to the Author):

The authors have addressed my concerns, and I look forward to seeing the publication in print.

Reviewer #3 (Remarks to the Author):

No further comments